# From gauging to duality in one-dimensional quantum lattice models

Bram Vancraeynest-De Cuiper[1,*], José Garre-Rubio[2,3], Frank Verstraete[1,4], Kevin Vervoort[1], Dominic J. Williamson[5], and Laurens Lootens[4]

[1] Department of Physics and Astronomy, Ghent University, Krijgslaan 299, 9000 Gent, Belgium
[2] University of Vienna, Faculty of Mathematics, Oskar-Morgenstern-Platz 1, 1090 Vienna, Austria
[3] Instituto de Física Teórica, UAM-CSIC, C. Nicolás Cabrera 13-15, Cantoblanco, 28049 Madrid, Spain
[4] Department of Applied Mathematics and Theoretical Physics, University of Cambridge, Wilberforce Road, Cambridge, CB3 0WA, United Kingdom
[5] School of Physics, The University of Sydney, NSW 2006, Australia

* bram.vancraeynestdecuiper@ugent.be

**Abstract**

Gauging and duality transformations, two of the most useful tools in many-body physics, are shown to be equivalent up to constant depth quantum circuits in the case of one-dimensional quantum lattice models. This is demonstrated by making use of matrix product operators, which provide the lattice representation theory for global (categorical) symmetries as well as a classification of duality transformations. Our construction makes the symmetries of the gauged theory manifest and clarifies how to deal with static background fields when gauging generalised symmetries.

# Introduction

Recent years have witnessed an increased interest in generalised notions of finite global symmetry in quantum theories [GKSW14, FMT22]. These generalisations encompass in particular *non-invertible* symmetries in which the algebraic structure governing the symmetry and its 't Hooft anomaly is no longer a group, but instead a *(higher) fusion category* [Ost01, FRS02, TW19, Sha23, SN23]. Such symmetries occur ubiquitously in both the continuum [FRS02, GKSW14, TW19, KORS20] as well as in lattice models [HEH+15, AMF16, VBW+18, VLTV21, MABT23, LDOV23, SSS24, SS24, LSY24]. The focus of this manuscript is on fusion categorical symmetries in one-dimensional lattice models.

Based on tensor network representations of string-net ground state models [VWPGC06, BAV09, cWB+14, BMW+15, LFH+20] – which are a particular case of the Turaev-Viro-Barrett-Westbury state-sum construction with symmetry category $\mathcal{C}$ as input – explicit *matrix product operator* (MPO) representations of fusion categorical symmetries have been classified and realized [BMW+15, WBV17, VBW+18, SSS24, SS24, CAW24]. Succinctly, for a categorical symmetry $\mathcal{C}$ each choice of *module category* $\mathcal{R}$ over it provides an explicit MPO representation of $\mathcal{C}$ [Ost01, EGNO15]. Subsequently, for the chosen module category, local operators that commute with the categorical symmetry can be built from *generalised Clebsch-Gordan coefficients* [BLV22], thereby providing a systematic machinery to construct local lattice Hamiltonians with a certain categorical symmetry. In particular, the *algebra of local symmetric operators* [CON11, Jon23, JSW24] is encapsulated in the *Morita dual* fusion category $\mathcal{C}^{\star}_{\mathcal{R}}$, and is thus fixed by the choice of symmetry and its representation. Generically these MPOs and local Hamiltonians are represented on Hilbert spaces without local tensor product structure, but which rather obey local constraints [FTL+06, BG17, AFM20].

Given such a $\mathcal{C}$-symmetric model, it was demonstrated in a recent series of papers [LDOV23, LDV24], how different choices of module categories $\mathcal{M}$ over $\mathcal{C}$ lead to distinct *dual* theories. Notable examples of such exact lattice dualities include Kramers-Wannier, Jordan-Wigner and Kennedy-Tasaki duality [KW41, JW28, KT92, Osh92]. Under this operation the symmetry of the dual model is given by the Morita dual of $\mathcal{C}$ with respect to $\mathcal{M}$. Physically, these categorical dualities are characterized by the fact that *local symmetric* operators are mapped to *dual local* operators which commute with the dual symmetry and *charged local* operators are mapped to *string operators* [CON11, MMT22]. From the skeletal data of $\mathcal{M}$, MPOs can be constructed which intertwine the dual models.

On the other hand, it is known that gaugings of a two-dimensional $\mathcal{C}$-symmetric model are given by *special haploid symmetric Frobenius algebra objects* $A$ in $\mathcal{C}$ [Ost01, FRS02, Sch13]. Such an algebra object encodes both a collection of symmetry lines that are gauged as well as a way to cancel the anomaly, or *discrete torsion*. On the level of the partition function the gauged model is obtained from the ungauged partition function by inserting a network of lines labelled by $A$ on a dual triangulation of the underlying manifold [CR12, BT17, Tac17, DLWW23]. The defining properties of $A$ then guarantee invariance of the gauged partition functions under recouplings of the $A$ defects. In this setting it is understood that the topological defects of the gauged model correspond to *bimodules* over $A$, encoded in the fusion category $\mathsf{Bimod}_{\mathcal{C}}(A)$ [FRS02, BT17]. Notably, the gauging is not determined by $A$ itself, but rather by its *Morita class*. Indeed, given $A$, the category $\mathsf{Mod}_{\mathcal{C}}(A)$ of $A$-modules can be constructed, which constitutes a module category over $\mathcal{C}$. As such, different algebra objects can produce the same category of modules, which ultimately dictates the gauging.

In this work, we make this connection between dualities and gauging of categorical symmetries via algebra objects explicit. Given a symmetry $\mathcal{C}$ and MPO representation specified by $\mathcal{R}$, we construct a *gauging map* following the approach taken in refs. [HVAS+15, SSY25]. More precisely, from a chosen $A$ we construct a local *Gauss law*, from which in turn a global projector onto the gauge-invariant subspace is made. This point-of-view provides an interpretation of the defects contained in $A$ as being the *gauge degrees of freedom* for the symmetry, in accordance with ref. [SSY25, FRBC25]. We then proceed by showing that the gauging map is equivalent to an MPO duality operator of ref. [LDOV23, LDV24] by means of a constant-depth unitary quantum circuit. This circuit maps the gauging map into an *effective Hilbert space* characterized by local constraints encoded in the module category $\mathcal{R}'$ over $\mathcal{C}^{\star}_{\mathcal{R}}$. As detailed in the main text, $\mathcal{R}'$ is fully specified by the choice of $\mathcal{M}$ and $\mathcal{R}$. As such, the dual

global symmetry is manifestly given by $\mathcal{C}^{\star}_{\mathcal{M}}$, which is equivalent to $\mathsf{Bimod}_{\mathcal{C}}(A)$ [BT17, LDWV23]. The procedure can be summarized in a triangle diagram:

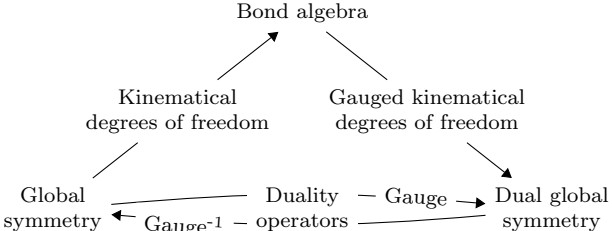

This diagram commutes in the sense that fixing the module category $\mathcal{R}$ representing the global symmetry $\mathcal{C}$ on the lattice – corresponding to the left side of the triangle – and a duality $\mathcal{M}$ – the bottom line – the dual symmetry and its lattice representation is fixed. Also note that the bond algebra of local symmetric operators, encoded in $\mathcal{C}^{\star}_{\mathcal{R}}$, is shared by both the ungauged and the gauged model.

The dualities of ref. [LDOV23, LDV24] are isometries only when the Hilbert space is supplemented with an extra degree of freedom representing the *symmetry-twisted boundary condition*. In the presence of such twists, the Hilbert space decomposes in *topological sectors* which are permuted under the duality mappings [PZ00, Ben14, Rad18, LOZ23]. We also provide a version of our gauging map in the presence of such a twist and demonstrate how it maps to a collection of duality *tubes*.

Our results can be extended in a number of ways. For one, the algebra objects considered in this manuscript admit higher-dimensional generalisations. In particular for the case of two spatial dimensions, it is expected that higher-dimensional generalisations of the gauging map [HVAS+15, MABT23, VDCGR24] can be related to the previously constructed duality maps of ref. [DT23] via a circuit generalising the one presented here. On the other hand, throughout the manuscript we restrict to internal symmetries. A natural question is how spacetime symmetries such as reflection or time-reversal symmetry fit in the framework. This would require addressing the notion of orientation-reversing domain walls and their Gauss law. We also expect that in this case the tensor network formalism will provide insight, given existing partial progress in this direction [CV14, JR15, VDCBD+22].

**Organisation of the manuscript:** We begin sec. 2 by reviewing generic properties of the MPO symmetries considered in this manuscript, before proceeding to the construction of the gauging map from a choice of algebra object. To this end we define a generalised commuting Gauss law from which a global projector onto the gauge-invariant subspace is defined. In sec. 3 we review the construction of duality MPOs and show the unitary equivalence with the gauging map in sec. 4. In section 5 we comment on the case of full gaugeability and how this implies the existence of a short-range entangled symmetric state, and the absence of an anomaly. We then provide a gauging map acting on symmetry-twisted boundary conditions in sec. 6. Algebra objects and their corresponding module categories for the category of representations of the symmetric group $S_3$ and the Haagerup fusion categories are presented in sec. 7. A review of the relevant category theory used throughout the manuscript is relegated to the appendix. A guide to our notation is provided in the following table.

| Physical notion | Categorical notion | Notation |
|---|---|---|
| Global symmetry ungauged model | Fusion category | $\mathcal{C}$ |
| Kinematical degrees of freedom | $\mathcal{C}$-module category | $\mathcal{R}$ |
| Algebra symmetric operators | Morita dual fusion category | $\mathcal{C}^{\star}_{\mathcal{R}}$ |
| Gaugeable subsymmetry | Frobenius algebra in $\mathcal{C}$ | $A$ |
| Duality operators | $\mathcal{C}$-module category | $\mathcal{M} \simeq \mathsf{Mod}_{\mathcal{C}}(A)$ |
| Gauged kinematical degrees of freedom | $\mathcal{C}^{\star}_{\mathcal{M}}$-module category | $\mathcal{R}'$ |
| Global symmetry gauged model | Morita dual fusion category | $\mathcal{C}^{\star}_{\mathcal{M}} \simeq \mathsf{Bimod}_{\mathcal{C}}(A)$ |

# Gauging procedure for algebra objects

*We construct a gauging map for any finite symmetry described by a unitary fusion category $\mathcal{C}$ from a choice of haploid symmetric separable Frobenius algebra object in $\mathcal{C}$. Such an algebra object fully characterizes an anomaly-free subset of symmetries in $\mathcal{C}$ together with a choice of generalised discrete torsion. This gauging map directly generalises the one from ref. [HVAS+15], which was formulated for an anomaly-free invertible symmetry.*

In this manuscript we are concerned with generalised finite symmetries described by *unitary fusion categories* (UFCs). It was shown in ref. [LFH+20] that any UFC admits a faithful and internal matrix product operator (MPO) representation of finite bond dimension provided an extra collection of data encoded in a choice of *(left) module category* over the symmetry category. Since the proposed generalised gauging procedure is oblivious to the concrete lattice realization of the symmetry, we first review the main properties of the symmetry MPOs before turning to the construction of the gauging map.

Throughout this manuscript we rely heavily on the graphical calculus for tensor networks in which a generic MPO tensor is depicted as

$$\vcenter{\hbox{\includegraphics{eq1left}}} \equiv \sum_{\substack{i,j \\ \alpha,\beta}} \vcenter{\hbox{\includegraphics{eq1right}}} \; |\alpha\rangle\langle\beta| \otimes |i\rangle\langle j| \,. \tag{1}$$

We refer to $i, j$ and $\alpha, \beta$ in the above equation as *physical* and *virtual indices* respectively, borrowing standard tensor network language. All vector spaces we consider are finite-dimensional. In this depiction, and below, the label $X$ is used to both denote the MPO tensor and the virtual space. A uniform and periodic MPO can then be constructed by concatenating individual MPO tensors and contracting basis vectors on the virtual level. In the presence of symmetry twists the symmetry operators have to be modified in a way that is explained below in sec. 6. Graphically, the periodic matrix product operators are

$$\vcenter{\hbox{\includegraphics{eq2}}} , \tag{2}$$

with the convention that the left and rightmost virtual indices are identified and summed over so as to implement the periodic boundary condition. Note that the (closed) MPOs are invariant under gauge transformations on the virtual level. From now on we assume without loss of generality that, potentially after carrying out such a gauge transformation, the symmetry MPOs are block-injective, meaning that the MPO tensors are block diagonal and each block generates an injective symmetry MPO [HV16]. At the level of the virtual spaces, this boils down to a block decomposition of the form $X \simeq \bigoplus_i N_{X_i}^X X_i$, where some of the blocks may appear multiple times given by the integer $N_{X_i}^X$, and the total number of distinct blocks is finite. Furthermore we assume that the MPOs are closed under composition. Concretely, this means that for any triple of blocks $X_1, X_2, X_3$ there exists an integer denoted by $N_{X_1 X_2}^{X_3}$, referred to as *N-symbol*, such that independent of system size one has:

$$\vcenter{\hbox{\includegraphics{eq3left}}} = \sum_{X_3} N_{X_1 X_2}^{X_3} \vcenter{\hbox{\includegraphics{eq3right}}} . \tag{3}$$

Herein, the sum is interpreted so as to run over all injective blocks with the convention that $N_{X_1 X_2}^{X_3} = 0$ whenever $X_3$ does not appear in the block decomposition of $X_1 \otimes X_2$. Among the injective blocks there is a distinguished one labelled by $\mathbb{1}$ such that $N_{X_1 \mathbb{1}}^{X_2} = N_{\mathbb{1} X_1}^{X_2} = \delta_{X_1, X_2}$ for all $X_1, X_2$. As such, $N$ defines a *fusion structure* on the collection of blocks $\{X_i\}_i$. Further requiring that (3) holds for all boundary conditions, the fundamental theorem of MPOs implies there exists an intertwining *fusion tensor* that implements the above decomposition on the level of the individual MPO tensors [BMW$^+$15]. Graphically, this intertwining property can be depicted as

$$ (4) $$

Herein, $i = 1, ..., N_{X_1 X_2}^{X_3}$ labels the $N_{X_1 X_2}^{X_3}$ linearly independent fusion tensors and corresponds to the multiplicity of the block $X_3$ in the decomposition of $X_1$ and $X_2$. We are also guaranteed the existence of a corresponding right inverse, or *splitting tensor*, satisfying the orthogonality condition

$$ = \delta_{X_3, X_3'} \delta_{i, i'} \, \mathrm{id}_{X_3}, \qquad (5) $$

so that the multiplicity indices $i$ and $i'$ label dual orthogonal bases for the *fusion* and *splitting space* situated on the corresponding tensors, as well as a completeness condition spelled out in [BMW$^+$15]. As demonstrated therein, associativity of the multiplication of three injective MPOs requires the existence of a set of invertible matrices, henceforth called *F-symbols*, that encode the two distinct ways in which three injective MPOs can be recoupled. Graphically this statement boils down to the recoupling identity

$$ = \sum_{X_6, k, l} \left( F_{X_4}^{X_1 X_2 X_3} \right)_{X_5, ij}^{X_6, kl} \qquad \qquad , \qquad (6) $$

which has to be satisfied for every choice of the external labels $X_1, X_2, X_3$ and $X_5, i, j$. Consistency of the recoupling of four MPOs reveals that the F-symbols need to satisfy a coherence condition known as the *pentagon equation*, which in turn guarantees that every allowed recoupling of fusion tensors is consistent. Crucially, for the symmetries considered in this manuscript, there exists an adequate basis for the junction spaces in eq. (6) in which all F-symbols are unitary as matrices in the sense that

$$ \sum_{X_6, k, l} \left( F_{X_4}^{X_1 X_2 X_3} \right)_{X_5, ij}^{X_6, kl} \left( \bar{F}_{X_4}^{X_1 X_2 X_3} \right)_{X_5', i'j'}^{X_6, kl} = \delta_{X_5, X_5'} \delta_{i, i'} \delta_{j, j'}, \qquad (7) $$

where the overline denotes complex conjugation. Up to basis transformations on the Hom spaces, it turns out that the number of solutions to the pentagon equation is finite for any valid fusion structure as specified by an N-symbol.

The set of labels $\{X_i\}_i$, assuming the existence of a unique trivial label $\mathbb{1}$ and the notion of duals, endowed with a fusion structure and a unitary solution to the pentagon equation exactly constitutes the defining data of a *unitary fusion category* – conventionally denoted by $\mathcal{C}$ – as demonstrated in refs. [BMW$^+$15, WBV17, LFH$^+$20]. Some relevant notions from the theory of fusion categories and the notation we use are summarized in section A. In this sense, the MPOs considered in this paper form explicit finite-dimensional lattice representations of the (abstract) graphical calculus of fusion

category theory, cfr. eq. (6). In particular this means that we can identify the collection of injective blocks with the set of (representatives of isomorphism classes of) simple objects of $\mathcal{C}$, denoted by $\mathcal{I}_{\mathcal{C}}$, and the F-symbols appearing in the recoupling identity eq. (6) coincide with the matrix elements of the components of the monoidal associator of $\mathcal{C}$, as defined per eq. (A.1). We now proceed to discuss gauging (part of) the $\mathcal{C}$ symmetry.

Following the paradigm of ref. [HVAS+15], gauging happens by introducing new gauge degrees of freedom between physical sites and enforcing a projector onto the gauge-invariant subspace of this enlarged Hilbert space. This projector is constructed as the product of local commuting projectors that represent a Gauss law. In the invertible symmetry case covered in ref. [HVAS+15] introducing the gauge degrees of freedom boils down to tensoring the initial – 'matter' – Hilbert space with copies of the group algebra $\mathbb{C}[G]$ localized on the links of the lattice. Subsequently, the gauge degrees of freedom are initialized in a configuration which can then be absorbed in the global projection, so as to end up with a gauging map acting on the initial Hilbert space.

As anticipated in the introduction, a choice of *special haploid symmetric Frobenius algebra object* in $\mathcal{C}$ – henceforth often referred to as simply (Frobenius) algebra object – exactly encodes the data needed to generalise the gauging procedure of ref. [HVAS+15] to fusion-categorical symmetries. Recall that loosely speaking a Frobenius algebra object in $\mathcal{C}$ is a (generically non-simple) object $A \simeq \bigoplus_{X \in \mathcal{I}_{\mathcal{C}}} N_X^A X$ together with multiplication and comultiplication maps, denoted by $\mu : A \otimes A \to A$ and $\Delta : A \to A \otimes A$ respectively, which are (co)associative on the nose and satisfy a number of further criteria spelled out in section A. In particular, a finite symmetry $\mathcal{C}$ is called *(fully) gaugeable* when the quantum dimensions of all simple objects are integer and the object $\bigoplus_{X \in \mathcal{I}_{\mathcal{C}}} d_X X$ can be endowed with a Frobenius structure. Physically, as an object in $\mathcal{C}$, $A$ exactly encodes the symmetries that are being gauged while the Frobenius structure on $A$ encodes a particular way of gauging them, thereby providing a more general notion of *discrete torsion*. An obstacle arising in the general setting is that the symmetries $\mathcal{C}$ are typically represented on Hilbert spaces that are not naturally equipped with a tensor product structure. Hence, it is not immediately apparent what the appropriate notion of a gauge degree of freedom is in this case. This issue is remedied in the following section where we demonstrate how symmetry twists labelled by $A$ can be inserted in the non-tensor product Hilbert space and how these defects play the role of gauge field in this setting.

In the remainder of this section we will absorb the trivial gauge field directly into the gauging map to obtain the gauging map that acts on the original Hilbert space. Given a Frobenius algebra $A$ in $\mathcal{C}$, we now proceed by proposing a generalised notion of Gauss law [GRK22]. Drawing upon our graphical calculus, we consider the following tensor network operator:

$$\mathcal{P} := \qquad (8)$$

In this we have defined

$$\qquad (9)$$

and the trivalent tensors are constructed from the fusion and splitting tensors introduced above, with

the components of $\mu$ and $\Delta$ defined per. eq. (A.9) as follows:

$$
\mu := \sum_{\substack{X_1,X_2,X_3 \\ i_1,i_2,i_3,l}} \mu^{(X_1,i_1),(X_2,i_2)}_{(X_3,i_3),l} \quad , \quad \Delta := \sum_{\substack{X_1,X_2,X_3 \\ i_1,i_2,i_3,l}} \Delta^{(X_3,i_3),l}_{(X_1,i_1),(X_2,i_2)} \quad . \tag{10}
$$

In this definition $i_1, i_2, i_3$ stand for the multiplicity of $X_1, X_2, X_3$ in $A$ respectively, and take values in $i_j = 1, 2, ..., N^A_{X_j}$, $j = 1, 2, 3$. The operator (8) should be interpreted as acting on a physical site and two neighboring gauge fields taking values in the object $A$. As such, it can be thought of as a localized version of the symmetry MPO (2) restricted to the subset of symmetries encoded in $A$.

We now show that the $\mathcal{P}$'s are (local) mutually commuting projection operators. Whenever two of these operators overlap in a common gauge degree of freedom, their commutation is graphically depicted as

$$
\overset{(A.8)}{=} \tag{11}
$$

In this figure and those that follow we suppressed $A$ labels when they can be inferred from the context. This commutation property can be directly deduced from the first Frobenius condition (A.8) which is satisfied by the multiplication and comultiplication maps. The fact that $\mathcal{P}$ is a projector follows in turn from the (co)associativity of $\mu$ and $\Delta$ together with the intertwining property (4) and the normalization of the Frobenius algebra (A.7):

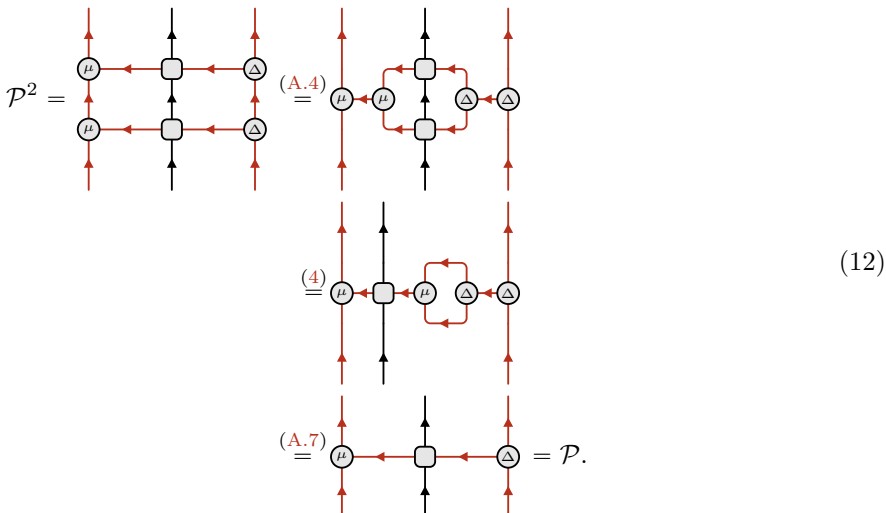

$$
\tag{12}
$$

The fact that these projectors mutually commute allows us to unambiguously define the following (global) projection operator on periodic boundary conditions:

$$
\tag{13}
$$

In order to define a gauging map acting on the original Hilbert space as anticipated we choose a trivial initial gauge configuration which is acted upon by the gauging map. Such a trivial gauge configuration is provided by the unit structure of the Frobenius algebra. Indeed, the unit provides a vector, denoted

by $|\mathbb{1}_A\rangle \in \mathcal{C}(\mathbb{1}, A)$, such that

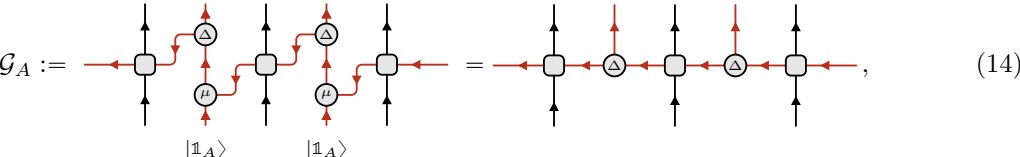

$$\mathcal{G}_A := \qquad\qquad = \qquad\qquad , \tag{14}$$

is exactly the sought-after gauging map. The action of the gauging map $\mathcal{G}_A$ on a local operator $O$ that commutes with the symmetry MPOs eq. (2) is then given via the prescription

$$O_g \cdot \mathcal{G}_A = \mathcal{G}_A \cdot O, \tag{15}$$

with $O_g$ the local gauged operator.

Despite being explicit, the gauging map (14) suffers from a number of shortcomings. Most notably, it is not obvious from its presentation in eq. (14) that the global symmetries of the gauged model are encoded in the fusion category $\mathsf{Bimod}_\mathcal{C}(A)$ of A-A-bimodules in $\mathcal{C}$, as argued in ref. [BT17]. This is particularly troublesome when trying to consider symmetry-twisted boundary conditions in the gauged model, which is necessary to obtain an invertible version of the gauging map. Moreover, gauging maps constructed from *Morita equivalent* Frobenius algebras should be considered physically equivalent even though their MPO representations (14) are distinct. These weaknesses are remedied by considering a basis in which the above gauging map boils down to the duality matrix product operators constructed in ref. [LDOV23, LDV24] based on the MPO representations of bimodule categories of ref. [LFH+20]. In the following section we review the duality matrix product operators pioneered in ref. [LDOV23, LDV24] before demonstrating the unitary equivalence with the gauging maps constructed above.

SECTION 3

### Duality matrix product operators

*Given a finite symmetry $\mathcal{C}$ we review the construction of ref. [LFH+20] of tensor network representations of the $\mathcal{C}$ symmetry operators, labelled by indecomposable module categories over $\mathcal{C}$, and the local operators invariant under them. We review the category theoretic approach to dualities proposed in ref. [LDOV23, LDV24] in which different lattice realizations of this local operator algebra amount to performing a duality. In preparation for the following section we revisit the matrix product operators intertwining the dual local operator algebras.*

---

From the results of ref. [BLV22] one can infer a generalised Wigner-Eckart theorem which states that local operators that commute with a finite symmetry $\mathcal{C}$ can be decomposed into *generalised Clebsch-Gordan coefficients*. Distinct collections of such coefficients are in one-to-one correspondence with distinct lattice realizations of the symmetry $\mathcal{C}$ and are classified by (finite indecomposable) *(left) module categories* $\mathcal{R}$ over $\mathcal{C}$, whose definition is revisited in section A.2. More precisely, these coefficients boil down to (the components of) the module associator of $\mathcal{R}$ as a (right) module category over the *Morita dual* fusion category $\mathcal{C}_\mathcal{R}^\star$. Recall that $\mathcal{C}_\mathcal{R}^\star$ is defined as $\mathsf{Fun}_\mathcal{C}(\mathcal{R}, \mathcal{R})$, i.e. the category of $\mathcal{C}$-module endofunctors of $\mathcal{R}$. As such, $\mathcal{R}$ is endowed with the structure of an *invertible $(\mathcal{C}, \mathcal{C}_\mathcal{R}^\star)$-bimodule category*. Below, we denote simple representative objects of $\mathcal{R}$ and $\mathcal{C}_\mathcal{R}^\star$ by $\{R_i\}_i$ and $\{Y_i\}_i$ respectively. The general framework of tensor network representations of bimodule categories, pioneered in ref. [LFH+20],

suggests the following trivalent *triple line* depiction of the generalised Clebsch-Gordan coefficients:

$$
\equiv \sum_{\{R_n\}_n} \sum_{i,j,l} \; \cdots \; |R_1 Y_1 R_3, i\rangle \otimes |R_3 Y_2 R_2, j\rangle \otimes \langle R_1 Y_3 R_2, l|
$$

$$
\equiv \sum_{\{R_n\}_n} \sum_{i,j,l} \big(F_{R_2}^{R_1 Y_1 Y_2}\big)_{R_3,ij}^{Y_3,kl} \; |R_1 Y_1 R_3, i\rangle \otimes |R_3 Y_2 R_2, j\rangle \otimes \langle R_1 Y_3 R_2, l|,
\tag{16}
$$

$$
\equiv \sum_{\{R_n\}_n} \sum_{i,j,l} \; \cdots \; |R_1 Y_3 R_2, l\rangle \otimes \langle R_1 Y_1 R_3, i| \otimes \langle R_3 Y_2 R_2, j|
$$

$$
\equiv \sum_{\{R_n\}_n} \sum_{i,j,l} \big(\bar{F}_{R_2}^{R_1 Y_1 Y_2}\big)_{R_3,ij}^{Y_3,kl} \; |R_1 Y_3 R_2, l\rangle \otimes \langle R_1 Y_1 R_3, i| \otimes \langle R_3 Y_2 R_2, j|.
$$

Throughout, we assume a basis for the Hom spaces in which these F-symbols are unitary when interpreted as linear maps $F_{R_2}^{R_1 Y_1 Y_2} : \bigoplus_{Y_3} \mathcal{C}_{\mathcal{M}}^{\star}(Y_1 \otimes Y_2, Y_3) \otimes \mathcal{R}(R_1 \lhd Y_3, R_2) \xrightarrow{\sim} \bigoplus_{R_3} \mathcal{R}(R_1 \lhd Y_1, R_3) \otimes \mathcal{R}(R_3 \lhd Y_2, R_2)$, and $i,j,k,l$ in the above denote basis vectors in these respective vector spaces and their duals. In eq. (16), and in similar expressions below, we adapt the graphical convention that unlabelled blue strands are summed over, and unlabelled gray patches stand for a sum over basis vectors, i.e.

$$
\equiv \sum_{\substack{R_1,R_2 \\ \in \mathcal{I}_{\mathcal{R}}}} \quad \equiv \sum_{\substack{R_1,R_2 \\ \in \mathcal{I}_{\mathcal{R}}}} \sum_i \quad |R_1 Y R_2, i\rangle,
$$

$$
\equiv \sum_{\substack{R_1,R_2 \\ \in \mathcal{I}_{\mathcal{R}}}} \quad \equiv \sum_{\substack{R_1,R_2 \\ \in \mathcal{I}_{\mathcal{R}}}} \sum_i \quad \langle R_1 Y R_2 i| .
\tag{17}
$$

Contraction of these triple line tensors takes place via identification of the simple objects on module strands of concatenated tensors, pairing basis vectors, and summing over them as detailed in ref. [LDOV23].

Under the above conventions, a local $\mathcal{C}$-symmetric two-site operator can be depicted as

$$
\mathbb{h}_{i,n}^{\mathcal{R}} := \sum_{\{Y_m\}_m} \sum_{i,j} \lambda_{i,n}(\{Y_m\}_m, i, j) \quad \cdots \quad ,
\tag{18}
$$

where $\{\lambda_{i,n}\}$ is a collection of complex numbers which may depend on all internal labels of the tensor on the right. For a generic choice of module category $\mathcal{R}$ the local operators (18) act on a Hilbert space which is inherently constrained, as enforced by the action of $\mathcal{C}_{\mathcal{R}}^{\star}$ on $\mathcal{R}$. In fact, the Hilbert space only has a tensor product structure when $\mathcal{R}$ contains only one simple object, and is hence equivalent to the category of (finite-dimensional) vector spaces. More formally, the Hilbert space can be written as

$$
\mathrm{Span}_{\mathbb{C}} \left\{ \quad \cdots \quad \Big| \{Y_{i+\frac{1}{2}}\}_i, \{R_i\}_i, \{i_{i+\frac{1}{2}}\}_i \right\},
\tag{19}
$$

where periodic boundary conditions are implemented by identifying $R_1 \equiv R_{L+1}$ for some chain length $L$. A generic symmetric local Hamiltonian acting on this space is then constructed as $\mathbb{H}^{\mathcal{R}} = \sum_i \sum_n \mathbb{h}_{i,n}^{\mathcal{R}}$. Note that one can always choose $\mathcal{R} = \mathcal{C}$, in which case the above Clebsch-Gordan coefficients boil down to the F-symbols of $\mathcal{C}$.[1] As such, the lattice models considered in refs. [LDOV23, LDV24] and this manuscript generalise the *anyonic spin chains* first considered in ref. [FTL$^+$06].

A *duality* in the sense of ref. [LDOV23] then amounts to changing the module category $\mathcal{R}$ while keeping the category $\mathcal{C}_{\mathcal{R}}^{\star}$ and the coefficients $\{\lambda_{i,n}\}$ fixed. Since $\mathcal{C}_{\mathcal{R}}^{\star}$ is kept fixed throughout, we move to a lightweight notation in the expressions below by writing $\mathcal{D}$ for $\mathcal{C}_{\mathcal{R}}^{\star}$, thereby suppressing the dependency of $\mathcal{D}$ on the initial symmetry and choice of module category. Under such a duality transformation the local symmetric operators (18) are mapped to dual local symmetric operators, charged local operators are mapped to disorder operators and the spectrum is preserved up to degeneracies if closed boundary conditions are accounted for as discussed below in sec. 6. The latter can heuristically be understood from the fact that the algebra of local operators, spanned by operators of the form (18), in particular its structure constants, is fully specified by the fixed fusion category $\mathcal{D}$.

Within the tensor network paradigm this change of module category can be implemented explicitly as a matrix product *intertwiner* that transmutes the dual local operators into each other. Indeed, for a generic duality between module categories $\mathcal{R}$ and $\mathcal{R}'$ the intertwiners are, by the result of ref. [LDOV23], labelled by $\mathcal{D}$-*module functors* between $\mathcal{R}$ and $\mathcal{R}'$ organized in $\mathsf{Fun}_{\mathcal{D}}(\mathcal{R}, \mathcal{R}')$, which itself is a *right* module category $\mathcal{M}$ over $\mathcal{C}$. To a given simple object $X \in \mathcal{M}$ there corresponds for every $R_1 \in \mathcal{I}_{\mathcal{R}}$, $Y \in \mathcal{I}_{\mathcal{D}}$ and $R_2' \in \mathcal{I}_{\mathcal{R}'}$ a matrix $^X\omega_{R_2'}^{R_1 Y}$ defined in eq. (A.13), from which we construct an MPO tensor whose components are

$$
\begin{array}{c}
R_1' \; \uparrow^{\,j} \; R_2' \\
i \longleftarrow X \!-\! l \\
\;\; R_1 \; |_{\,k}^{Y} \; R_2
\end{array}
\;:=\; \bigl({}^X\omega_{R_2'}^{R_1 Y}\bigr)_{R_1',ij}^{R_2,kl}.
\tag{20}
$$

Without loss of generality $^X\omega_{R_2'}^{R_1 Y}$ is assumed to be a unitary map $^X\omega_{R_2'}^{R_1 Y} : \bigoplus_{R_2} \mathcal{R}(R_1 \triangleleft Y, R_2) \otimes \mathcal{R}'(X \triangleright R_2, R_2') \xrightarrow{\sim} \bigoplus_{R_1'} \mathcal{R}'(X \triangleright R_1, R_1') \otimes \mathcal{R}'(R_1' \triangleleft Y, R_2')$. Notably, $^X\omega$ satisfies a consistency equation involving the F-symbols appearing in eq. (16) for the module categories $\mathcal{R}$ and $\mathcal{R}'$ which can be graphically depicted as the following tensor equality:

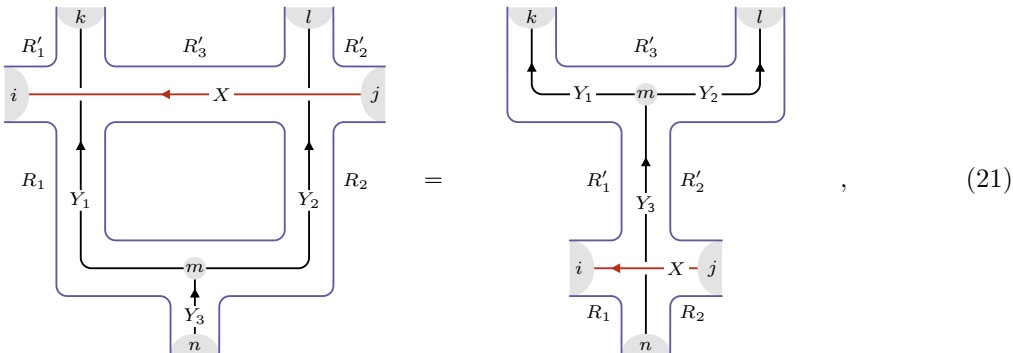

$$\tag{21}$$

for all valid labellings of the objects and multiplicity indices. A similar property is satisfied by the second tensor in eq. (16). This equality demonstrates that the MPO constructed from the tensor (20) labelled by the object $X \in \mathcal{M}$ transmutes local operators of one model corresponding to the choice of module category $\mathcal{R}$ to the one of the model based on $\mathcal{R}'$, thereby implementing the duality

---

[1]The term generalised Clebsch-Gordan coefficients stems from the fact that for the case of an invertible symmetry as described by the fusion category $\mathcal{C} = \mathsf{Vec}_G$, the category of $G$-graded vector spaces, the F-symbols appearing in (16) coincide with the usual Clebsch-Gordan coefficients when the choice $\mathcal{R} = \mathsf{Vec}$ is made. Here, $\mathsf{Vec}$ stands for the category of complex finite-dimensional vector spaces. In that case the Morita dual is found to be $(\mathsf{Vec}_G)_{\mathsf{Vec}}^{\star} \simeq \mathsf{Rep}(G)$, the fusion category of complex finite-dimensional representations of $G$ [EGNO15, LFH$^+$20].

transformation as foreseen. Importantly, choosing both module categories in eq. (20) equal to $\mathcal{R}$ exactly realizes the lattice representation of the symmetry operators encoded in $\mathcal{C}$, here thought of as $\mathcal{C} \equiv \mathsf{Fun}_{\mathcal{D}}(\mathcal{R}, \mathcal{R}) = \mathcal{D}_{\mathcal{R}}^{\star}$. In that instance it is more precise to refer to (20) as the *bimodule associator* of $\mathcal{R}$ as $(\mathcal{C}, \mathcal{D})$-bimodule category. Choosing both module categories equal to $\mathcal{R}'$ on the other hand reveals that the global symmetries of the dual model are encoded in the fusion category $\mathcal{D}_{\mathcal{R}'}^{\star}$, equivalently $\mathcal{C}_{\mathcal{M}}^{\star}$. The global symmetries of the gauged model are thus encoded in the Morita dual of the symmetry category of the ungauged model with respect to the category of duality operators. Note that all duality operators labelled by module functors in $\mathcal{M}$ implement physically the same duality operation. Being itself an invertible $(\mathcal{C}_{\mathcal{M}}^{\star}, \mathcal{C})$-bimodule category, $\mathcal{M}$ can be acted upon from the *right* by $\mathcal{C}$ and from the *left* by $\mathcal{C}_{\mathcal{M}}^{\star}$. As such, distinct duality operators can be converted into each other by pre- or postcomposing with symmetry operators in $\mathcal{C}$, respectively $\mathcal{C}_{\mathcal{M}}^{\star}$.

Given such $\mathcal{C}$ symmetry MPOs, what are the fusion and splitting tensors employed in the construction of the gauging map proposed in the previous section? Akin to the definition (16) we can consider tensors evaluating to the components of the *left* module associators of $\mathcal{R}$ over $\mathcal{C}$:

$$
\left( F_{R_3}^{X_1 X_2 R_1} \right)_{X_3, ij}^{R_2, kl} \quad \text{and} \quad \left( \bar{F}_{R_3}^{X_1 X_2 R_1} \right)_{X_3, ij}^{R_2, kl}. \tag{22}
$$

Again by virtue of two mixed pentagon equations these fusion tensors can be shown to satisfy the intertwining property (4) for the symmetry MPO tensors (20) and the recoupling identity (6) involving the $\mathcal{C}$ F-symbol, as showcased in ref. [LFH$^{+}$20]. As such, given an algebra $A$ in $\mathcal{C}$, explicit lattice representations of the multiplication and comultiplication maps read:

$$
:= \sum_{\substack{X_1, X_2, X_3 \\ i_1, i_2, i_3, l}} \mu_{(X_3, i_3), l}^{(X_1, i_1), (X_2, i_2)} \quad ,
$$

$$
:= \sum_{\substack{X_1, X_2, X_3 \\ i_1, i_2, i_3, l}} \Delta_{(X_1, i_1), (X_2, i_2)}^{(X_3, i_3), l} \quad ,
$$

(23)

where the sums should be interpreted as explained surrounding eq. (10). Collecting everything, for any choice of module category over the symmetry category $\mathcal{C}$ one finds a corresponding MPO representation of that symmetry so that for any algebra $A$ in $\mathcal{C}$ the generalised Gauss operator is given by:

$$
. \tag{24}
$$

This Gauss operator acts on the Hilbert space (19) supplemented with symmetry twists labelled by the object $A$. Following ref. [LDOV23] states in this enlarged Hilbert space can be depicted as

$$
, \tag{25}
$$

– 11 –

where the span is now taken over additional module labels $\widetilde{R}_i$. Note that since we have assumed $A$ to be haploid, meaning that the decomposition of $A$ contains a single copy of the monoidal unit $\mathbb{1}_{\mathcal{C}}$, there is a canonical way in which a basis state in the initial Hilbert space (19) can be embedded in this enlarged Hilbert space, namely via

$$\tag{26}$$

Equivalently, in line with the previous section, we can incorporate this choice of gauge field directly into the gauging map so as to obtain a gauging map acting on the original Hilbert space (19). The resulting operator can be depicted as:

$$\mathbb{G}_A := \tag{27}$$

As such, this procedure provides an interpretation of the $A$ symmetry twists in the extended Hilbert space (25) as the 'gauge field' for the subsymmetry $\mathcal{C}$ which is being gauged, in accordance with ref. [SSY25]. Notably, the insertion of the defects in (25) is independent of the Frobenius structure on $A$, which only appears in the definition of the Gauss constraint. In that sense, the Frobenius structure can be thought of as generalising the notion of discrete torsion to the categorical setting, as anticipated above.

This can be compared to ref. [HVAS$^+$15] whose results are obtained as a specific instance of this formalism. Indeed, the results of ref. [HVAS$^+$15] are recovered by choosing $\mathcal{R} = \mathsf{Vec}$, the category of (complex finite-dimensional) vector spaces, as module category over the fusion category $\mathcal{C} = \mathsf{Vec}_G$ of $G$-graded vector spaces which describes an invertible $G$ symmetry. In that case the objects $\{Y_i\}_i$ label representations of $G$. Gauging the full $G$ symmetry then happens via choosing $A \simeq \mathbb{C}[G]$, here thought of as the regular representation of $G$. The Hilbert space (25) then effectively boils down to the initial 'matter' Hilbert space tensored with copies of the group algebra $\mathbb{C}[G]$ in between the physical sites. Furthermore, one could twist the gauging map by a choice of 2-cocycle, i.e. $[\psi] \in H^2(G, \mathrm{U}(1))$, by choosing $A$ to be the $\psi$-twisted group algebra $\mathbb{C}^\psi[G]$ which as a vector space is spanned by states $\{|g\rangle, g \in G\}$ with algebra multiplication given by $|g\rangle \cdot |h\rangle = \psi(g,h)|gh\rangle$ [LDOV23, LSY24, VDCD25].

In preparation for the following section, let us note that the trivalent tensors evaluating to the left module associators (22) are a special instance of a more general kind of tensors which encode the composition of module functors. Indeed, given $\mathcal{M}, \mathcal{N}, \mathcal{O}$, all right module categories over $\mathcal{D}$, the unitary F-symbols introduced in (A.14) and their conjugates can be organised in triple line tensors of the form

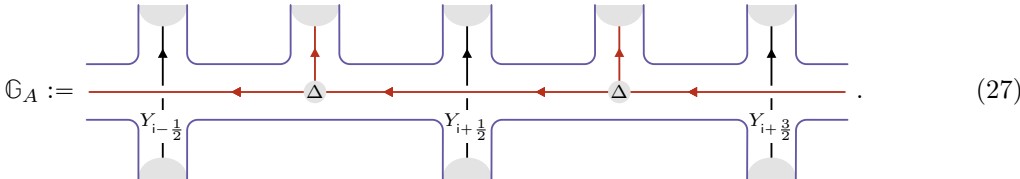

$$:= \left(F_O^{X_1 X_2 M}\right)_{X_3,ij}^{N,kl} \quad \text{and} \quad := \left(\bar{F}_O^{X_1 X_2 M}\right)_{X_3,ij}^{N,kl}, \tag{28}$$

where $M \in \mathcal{I}_{\mathcal{M}}$, $N \in \mathcal{I}_{\mathcal{N}}$, $O \in \mathcal{I}_{\mathcal{O}}$ and $X_1 \in \mathcal{I}_{\mathsf{Fun}_{\mathcal{D}}(\mathcal{N},\mathcal{O})}$, $X_2 \in \mathcal{I}_{\mathsf{Fun}_{\mathcal{D}}(\mathcal{M},\mathcal{N})}$, $X_3 \in \mathcal{I}_{\mathsf{Fun}_{\mathcal{D}}(\mathcal{M},\mathcal{O})}$. From $\mathcal{D}_{\mathcal{R}}^\star \equiv \mathcal{C}$ it follows that for the case where all three module categories are chosen to be equal to $\mathcal{R}$ these tensors indeed reduce to the matrix components of the left module associators of $\mathcal{R}$ over $\mathcal{C}$ as per eq. (22).

# Equivalence gauging map and duality operator

*In this section, we show how the gauging map (27) can be converted to a duality MPO from the previous section by means of a finite sequence of local unitary transformations, i.e. a constant depth quantum circuit.*

---

Given a finite symmetry $\mathcal{C}$ represented on the lattice via a choice of module category $\mathcal{R}$, how can the gauging map constructed from a Frobenius algebra object $A$ in $\mathcal{C}$ be related to a duality operator of the kind described in the previous section? By the results of refs. [Ost01, Sch13] there exists a right $\mathcal{C}$-module category such that $A$ corresponds to the *internal hom* of a particular simple module object. More precisely, this module category is exactly equivalent to the category $\mathcal{M} = \mathsf{Fun}_{\mathcal{D}}(\mathcal{R}, \mathcal{R}')$ of duality operators, for a particular $\mathcal{R}'$. Indeed, $\mathcal{R}'$ is determined by the fact that $\mathcal{M}$ is equivalent to the category $\mathsf{Mod}_{\mathcal{C}}(A)$ of *left $A$-modules* [Ost01, FRS02]. The internal hom reconstruction of $A$ is reviewed in section A.3. In what follows $X_A$ denotes the simple object such that $A \equiv \underline{\mathrm{Hom}}(X_A, X_A)$. The simple module object $X_A \in \mathcal{M}$ is precisely the one labelling the duality MPO to which the gauging map $\mathbb{G}_A$ (27) is unitarily equivalent by means of a unitary constant-depth quantum circuit, as we now demonstrate.

The internal hom reconstruction allows us to 'inflate' the comultiplication tensors (23) appearing in the gauging map (27) as follows [FRS03, Sch13]:

$$
\begin{array}{c} \text{(diagram)} \end{array} = \sum_{\substack{X_1, X_2, X_3 \\ i_1, i_2, i_3}} \frac{\sqrt{d_{X_1} d_{X_2} d_{X_3}}}{d_{X_A}^2} \quad \begin{array}{c} \text{(diagram)} \end{array} , \tag{29}
$$

where, as in eq. (22), the sum is over the simple objects appearing in the decomposition of $A$, and where in virtue of the defining property of the internal hom (A.15), $N_{X_j}^A = \dim_{\mathbb{C}} \mathcal{C}(X_j, A) = \dim_{\mathbb{C}} \mathcal{M}(X_A \triangleleft X_j, X_A)$ [Sch13]. Also remark that the innermost closed module loop on the right-hand side is valued in $\mathcal{R}'$ over whose simple objects we sum. Furthermore, the normalisation is chosen to match the one in eq. (A.16). In section C we demonstrate, based on our graphical calculus for triple line tensors, that the inflated representation of the comultiplication (29) satisfies the defining coalgebra property.

Before applying the aforementioned circuit, we first rewrite the gauging map by making use of eq. (29). We begin by inserting eq. (29) in the gauging map eq. (27), followed by using the orientation-reversing 'flags' introduced in section B to reverse some of the arrows in eq. (29). We then exploit the unitarity of the trivalent tensors, which in our graphical notation reads

$$
\sum_{X,i} \quad \begin{array}{c} \text{(diagram)} \end{array} = \delta_{R_1', R_2'} \delta_{k,k'} \delta_{l,l'} \quad \begin{array}{c} \text{(diagram)} \end{array} , \tag{30}
$$

so as to rewrite the gauging map as follows:

$$\mathbb{G}_A = \sum_{\substack{\{X_i\}_i \\ \{j_i\}_i}} d_{X_A}^{-\frac{L}{2}} \quad$$ 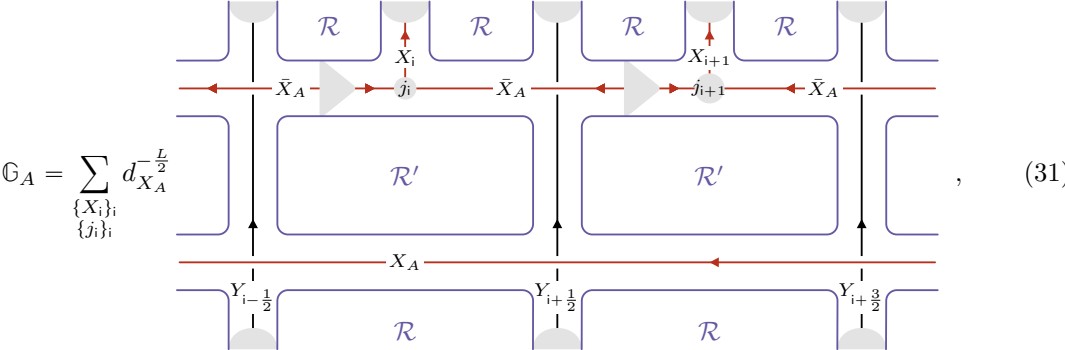 $$, \qquad (31)$$

where the letters $\mathcal{R}, \mathcal{R}'$ denote the module categories in which surrounding module strands are valued. Here and below, $\bar{X}_A \in \mathcal{I}_{\mathcal{M}^{\mathrm{op}}}$, where $\mathcal{M}^{\mathrm{op}} = \mathsf{Fun}_{\mathcal{D}}(\mathcal{R}', \mathcal{R})$, labels the unique 'opposite' duality strand specified by the fact that the composition of $\bar{X}_A$ and $X_A$ exactly yields the algebra object $A$ in $\mathcal{C}$.

Let us stress again that eq. (31) is an exact rewriting of the gauging map. In order to bring it into the form of a duality operator, we now invoke the following unitary gate defined in ref. [LDWV23] constructed from the fusion tensors eq. (28):

$$\text{(32)}$$

which should be interpreted as unitary between the Hilbert spaces $\bigoplus_{X,R_1,R_2} \mathcal{R}(X \triangleright R_2, R_1) \otimes \mathcal{C}(X, \bar{X}_A \otimes X_A) \xrightarrow{\sim} \bigoplus_{R_1,R_2,R'} \mathcal{M}(X_A \triangleright R_1, R') \otimes \mathcal{M}^{\mathrm{op}}(\bar{X}_A \triangleright R', R_2)$. Graphically, unitarity of this gate can be expressed by a rotated version of eq. (30).

Acting then with a layer of this gate on the strands labelled by objects $\{X_i\}_i$ in eq. (31), eq. (31) can be brought in the following form:

$$\mathbb{G}_A \cong d_{X_A}^{-\frac{L}{2}} \quad$$ 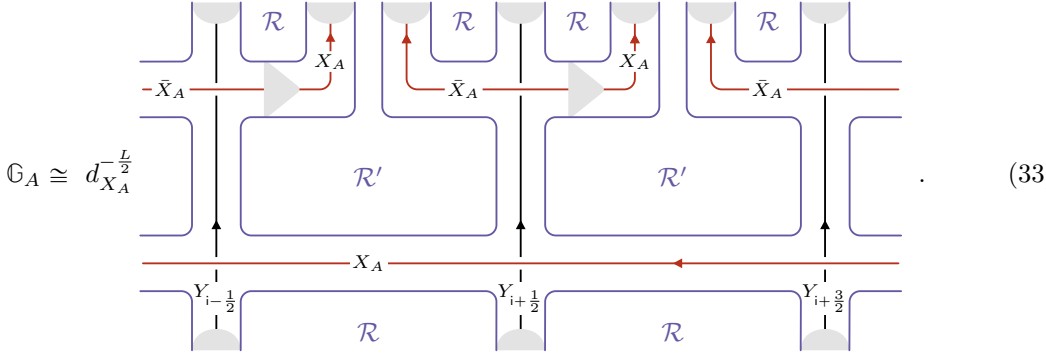 $$. \qquad (33)$$

At the bottom of eq. (33) we recognize the duality MPO implementing the duality $\mathcal{R} \to \mathcal{R}'$ labelled by $X_A$. The 'cups' on top can be removed by means of a local unitary transformation. To this end we also define the following unitary gate whose entries are expressed in terms of a rotated version of the $^{X_A}\omega$ tensors introduced in above eq. (20):

$$\text{(34)}$$

This gate can be used to thread the strands labelled by $X_A$ to the left through the physical degrees of freedom $\{Y_{i+\frac{1}{2}}\}_i$. Subsequently, by making use of the conjugate of the gate eq. (32) the $\bar{X}_A$ and $X_A$ strands can be fused together unitarily, in the process picking up an extra overall factor $d_{X_A}^{L/2}$. Due to the presence of the cup in eq. (33) the only allowed fusion outcome is $\mathbb{1}_{\mathcal{C}_{\mathcal{M}}^{\star}}$ which appears with multiplicity one in $X_A \otimes \bar{X}_A$. As such, one recovers eventually the duality MPO labelled by $X_A$ as anticipated:

$$\mathbb{G}_A \cong \mathbb{D}_{X_A} := \qquad \qquad \qquad \qquad \qquad \qquad \qquad \qquad \qquad \qquad \qquad \qquad . \qquad (35)$$

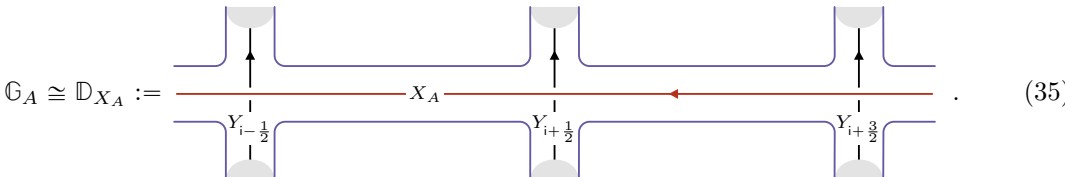

We commented earlier that the global symmetries of the dual model are encoded in the fusion category $\mathcal{C}_{\mathcal{M}}^{\star}$. In virtue of the fact the that the category $\mathcal{M}$ labelling the duality operators is equivalent to the category $\mathsf{Mod}_{\mathcal{C}}(A)$ of $A$-modules, $\mathcal{C}_{\mathcal{M}}^{\star}$ is indeed equivalent to $\mathsf{Bimod}_{\mathcal{C}}(A)$, in agreement with the results of refs. [FRS02, BT17]. This can be verified more explicitly by conjugating the symmetry MPOs labelled by objects in $\mathcal{C}_{\mathcal{M}}^{\star}$ with the circuit constructed above. Making use of the inflation trick eq. (29) it is then easy to verify that the resulting operator, labelled by some object in $\mathcal{C}$, indeed is endowed with the structure of an A-A bimodule in $\mathcal{C}$. More precisely, starting from the (non-simple) symmetry operator in the dual model labelled by $X_A \otimes X \otimes \bar{X}_A \in \mathcal{C}_{\mathcal{M}}^{\star}$, where $X \in \mathcal{C}$, we can conjugate it with the circuit to obtain the MPO corresponding to the object $A \otimes X \otimes A \in \mathcal{C}$. Interpreting this object as the *induced bimodule* $\mathrm{Ind}_{A|A}(X)$ defined in [FRS04, Def. 5.1], it then follows from [FRS04, Lemma 5.2], that every simple A-A bimodule is obtained in the decomposition of this MPO for a certain $X$.

We conclude this section with a few additional remarks. First, notice that the depth of the quantum circuit being constant and small is a direct consequence of the fact that the construction of the gauging map involves the introduction of an extensive number of defects labelled by $A$. As such it should be compared to the quantum circuit representation of the duality operator showcased in ref. [LDWV23], whose depth scales linearly in the system size in the generic case, and is constant in the nilpotent case when supplemented with measurements and feedforward operations.

A further natural question is whether every (simple) duality operator labelled by $X \in \mathcal{I}_{\mathsf{Fun}_{\mathcal{D}}(\mathcal{M},\mathcal{N})}$ is equivalent to a gauging map for a suitable choice of algebra object. The answer is affirmative and the algebra object is provided by the internal hom $\underline{\mathrm{Hom}}(X, X)$. In that case, the circuit that transmutes the gauging map $\mathbb{G}_{\underline{\mathrm{Hom}}(X,X)}$ to the duality MPO $\mathbb{D}_X$ exactly mimics the one above with $\underline{\mathrm{Hom}}(X, X)$ substituted for $A$ and $X$ for $X_A$.

As mentioned above, $\mathsf{Fun}_{\mathcal{D}}(\mathcal{M}, \mathcal{N})$ carries the structure of an invertible $(\mathcal{D}_{\mathcal{N}}^{\star}, \mathcal{C})$-bimodule category. In particular this implies that $\underline{\mathrm{Hom}}(X_A, X_A)$ can be interpreted as an internal hom object in the category $\mathcal{D}_{\mathcal{N}}^{\star}$, namely via $\mathsf{Fun}_{\mathcal{D}}(\mathcal{M}, \mathcal{N})(X \rhd X_A, X_A) \simeq \mathcal{D}_{\mathcal{N}}^{\star}(X, \underline{\mathrm{Hom}}(X_A, X_A))$. As such, $\underline{\mathrm{Hom}}(X_A, X_A)$ is also endowed with the structure of a Frobenius algebra in $\mathcal{D}_{\mathcal{N}}^{\star}$ which can be expressed in terms of the *left* $\mathsf{Fun}_{\mathcal{D}}(\mathcal{M}, \mathcal{N})$ module associators, akin to eq. (A.16). This algebra object thus also provides a gauging map which in this case turns out to be equivalent to the duality MPO $\mathbb{D}_{\bar{X}_A}$ implementing the duality $\mathcal{N} \to \mathcal{M}$. Note however that $\mathbb{G}_{\bar{X}_A} \circ \mathbb{G}_{X_A}$ is not equal to the identity operator on the periodic Hilbert space eq. (19). In fact, $\mathbb{G}_{\bar{X}_A} \circ \mathbb{G}_{X_A}$ yields the symmetry MPO labelled by the algebra object $A$, which is typically a non-invertible symmetry operator. It turns out that when the original and dual Hilbert spaces are supplemented with symmetry-twists, the duality MPOs can be implemented unitarily. This is the topic of section 6.

# Anomalies and short-range symmetric states

*We comment on the specific case where the symmetry is fully gaugeable and show that in that case we can think of gauging as averaging over a local version of the symmetry. We further argue that full gaugeability implies the existence of a short-range entangled $\mathcal{C}$-symmetric state, and therefore the absence of an anomaly.*

---------------

For invertible symmetries described by $\mathsf{Vec}_G^\omega$, the 3-cocycle $[\omega] \in H^3(G, U(1))$ poses an obstruction to gauging. Indeed, the possible algebra objects in $\mathsf{Vec}_G^\omega$ are $\psi$-twisted group algebras $\mathbb{C}^\psi[K]$, where $\omega|_K$ is trivial, $[\psi] \in H^2(K, U(1))$. Therefore, $\omega$ characterizes the anomaly of such a symmetry, and gauging with an algebra object $\mathbb{C}^\psi[K]$ amounts to $\psi$-twisted gauging of the non-anomalous subgroup $K$. As argued in [CLW11], the anomaly $\omega$ also provides an obstruction to the existence of a short-range entangled state that is symmetric, which is the generally accepted definition for an anomaly in $(1+1)$d.

For general non-invertible symmetries, the interpretation of the algebra object $A$ as a subset of symmetries that can be gauged is less clear. The reason for this is that in this case, one generally cannot think of the projectors in eq. (8) as an averaging over localised versions of the symmetry operators as in the invertible case. The exception to this is when the symmetry $\mathcal{C}$ is anomaly-free; in this case, it can be gauged completely, even if it is non-invertible. To see this, we rely again on the equivalence between algebra objects in $\mathcal{C}$ and internal Hom objects. Since gapped phases of a $\mathcal{C}$ symmetric theory are classified by module categories $\mathcal{P}$ over $\mathcal{C}$ [TW19, GRLM22], being anomaly-free implies the existence of $\mathsf{Vec}$ as a $\mathcal{C}$-module category, also referred to as a *fiber functor*. This requires $\mathcal{C}$ to be the representation category of a Hopf algebra $\mathcal{A}$, and we denote $\mathcal{C} = \mathsf{Rep}(\mathcal{A})$. The internal Hom construction provides an algebra object $A \equiv \underline{\mathrm{Hom}}(\mathbb{C}, \mathbb{C})$, with $\mathbb{C}$ the unique simple object in $\mathsf{Vec}$. As an object of $\mathsf{Rep}(\mathcal{A})$, $A$ is the regular representation, $A \simeq \oplus_{X \in \mathcal{I}_\mathcal{C}} d_X \cdot X$. It is this choice of algebra object that amounts to the complete gauging of the symmetry $\mathcal{C}$, as we show below.

Using the expression of the local Gauss constraint from section 2, we can write

$$
\mathcal{P} = \;\;\;\;\;\;\;\;\;\;\;\; = \sum_{a,i} \;\;\;\;\;\;\;\;\;\;\;\; =: \sum_a \frac{d_a}{\mathrm{FPdim}\mathcal{M}} \mathcal{P}^a,
\tag{36}
$$

where the triangular tensor is the projector of $A$ on the block labeled by $a$ and $i$ is the multiplicity of the simple object $a$ in $A$, cf. eq. (A.9). $\mathrm{FPdim}\mathcal{M} = \mathrm{FPdim}\mathcal{C} = \sum_{X \in \mathcal{I}_\mathcal{C}} d_X^2$ denotes the total quantum dimension of $\mathcal{M}$, which in our conventions is equal to that of $\mathcal{C}$. We can show using the graphical calculus and the explicit expressions of the multiplication and comultiplication maps that

$$
\mathcal{P}^a \mathcal{P}^b = \sum_c N_{ab}^c \mathcal{P}^c,
\tag{37}
$$

where the N symbol is that of the symmetry category $\mathcal{C}$. Indeed:

$$
\mathcal{P}^a \mathcal{P}^b \overset{\text{(A.4)}}{=} \frac{(\mathrm{FPdim}\mathcal{M})^2}{d_a d_b} \sum_{i,j} \;\;\;\;\;\;\;\;\;\;\;\;
$$

$$\stackrel{\text{(A.16)}}{=} \frac{(\text{FPdim}\mathcal{M})^2}{d_a d_b} \sum_{\substack{c,k,\\k',l}} \Big( \underbrace{\sum_{i,j} \mu^{(a,i),(b,j)}_{(c,k),l} \Delta^{(c,k'),l}_{(a,i),(b,j)}}_{\frac{d_a d_b}{d_c} \frac{1}{\text{FPdim}\mathcal{M}} \delta_{k,k'}} \Big)$$  (38)

$$\stackrel{\text{(7)}}{=} \sum_{c,k} N^c_{ab} \frac{\text{FPdim}\mathcal{M}}{d_c}$$  $$= \sum_c N^c_{ab} \mathcal{P}^c.$$

This demonstrates that the operators $\mathcal{P}^a$ are local versions of the (non-invertible) symmetry $\mathcal{C}$, and we can interpret the gauging by $A$ as gauging the full symmetry.

The equivalence between these gauging maps and duality operators from the previous section can now also be used to show that if a symmetry can be completely gauged, there must exist a symmetric short-range entangled state. Indeed, it suffices to consider the gauging map $\mathcal{G}_A$, which as we have shown corresponds to the duality operator labelled by $X_A$ up to local unitary transformations. Applied to the above setting, the gauging map for the algebra object $A$ in $\text{Rep}(\mathcal{A})$ is equivalent to the duality operator labelled by $\mathbb{C} \in \text{Vec}$. Acting with $\mathcal{G}_A^\dagger$ on a generic product state therefore produces an injective $\mathcal{C}$-symmetric MPS, since $\mathcal{G}_A$ is an injective MPO. This way of producing $\mathcal{C}$-symmetric MPS by ungauging the symmetry $\mathcal{C}$ lies at the heart of symmetric tensor network methods, as shown in [LDV25]. Note that the same procedure can also be applied to the setting where the symmetry category $\mathcal{C}$ contains an anomaly-free subcategory $\mathcal{C}' \subset \mathcal{C}$, despite $\mathcal{C}$ itself being anomalous. In that case one can, by choosing the appropriate algebra object, still construct a short-range entangled $\mathcal{C}'$-symmetric state. One notable example is the $\text{Vec}_{\mathbb{Z}_2}$ subcategory of $\text{Ising}$ [SSS24]; another is the $\text{Vec}_{\mathbb{Z}_3}$ subcategory of the Haagerup category $\mathcal{H}_3$, which is discussed in more detail below.

<p style="text-align:center">SECTION 6</p>

# Symmetry-twisted boundary conditions

*We propose a generalisation of the gauging map eq. (27) to act on a Hilbert space twisted by a symmetry defect. We demonstrate the equivalence with the symmetry-twisted duality operators introduced in ref. [LDV24].*

In the construction of the anyonic spin chains eq. (19), we identified the module labels $R_1 \equiv R_{L+1}$ so as to implement *periodic* boundary conditions. More generally, one can insert a *symmetry defect*, labelled by an object $B \in \mathcal{I}_\mathcal{C}$, before closing the chain. This results in a *symmetry-twisted* Hilbert space, given by

$$\text{Span}_\mathbb{C} \left\{ \text{ } \middle| \{Y_{i+\frac{1}{2}}\}_i, \{R_i\}_i, \{i_{i+\frac{1}{2}}\}_i \right\}.$$ (39)

The symmetry twist is *topological* in the sense that the Hilbert space obtained by moving the twist by one site is related to the original one by a unitary transformation given by the bimodule associator of $\mathcal{R}$. Note that for the choice $B = \mathbb{1}$ the periodic chain eq. (19) is retrieved.

Local operators acting on the twisted Hilbert space away from the twist are still given by ladder diagrams of the form eq. (18), whereas local operators that act across the defect are given by a modified version of eq. (18) in which a rotated version of the tensor eq. (20) with $\mathcal{R} = \mathcal{R}'$ is inserted, as detailed in ref. [LDV24]. As before, a duality amounts to changing the choice of module category $\mathcal{R} \to \mathcal{R}'$. The

duality MPOs defined above in eq. (35) now need to be promoted to *tubes*, given by matrix product operators of the form

$$
\mathfrak{T}^{B,B',X,X',k,k'}_{\mathcal{R}'|\mathcal{R}} :=
$$

(40)

where $X, X' \in \mathcal{I}_{\mathcal{M}}$, $B' \in \mathcal{I}_{\mathcal{C}^\star_{\mathcal{M}}}$ and $X'$ is contained in both $B \otimes X$ and $X \otimes B'$. The trivalent tensors used to fuse the twists with the duality strand are those defined in eq. (28) for the appropriate choice of module categories. Similar to the untwisted case, tubes of the type $\mathfrak{T}_{\mathcal{R}|\mathcal{R}}$ and $\mathfrak{T}_{\mathcal{R}'|\mathcal{R}'}$ commute with the aforementioned local operators so that they constitute the symmetry operators of the original and the dual model. Together, the collection of tubes $\{\mathfrak{T}_{\mathcal{R}|\mathcal{R}}, \mathfrak{T}_{\mathcal{R}|\mathcal{R}'}, \mathfrak{T}_{\mathcal{R}'|\mathcal{R}}, \mathfrak{T}_{\mathcal{R}'|\mathcal{R}'}\}$ is endowed with a multiplication defined via stacking of tubes as well as a dagger. As such they span a finite $C^*$-algebra [NY18, LVDCSV21, LDV24]. This algebra admits an Artin-Wedderburn decomposition in terms of simple matrix algebras which are identified with the *topological sectors* of the model. In particular, this algebra consists of two subalgebras spanned by tubes of the type $\mathfrak{T}_{\mathcal{R}|\mathcal{R}}$ and $\mathfrak{T}_{\mathcal{R}'|\mathcal{R}'}$ respectively, whose blocks are in one-to-one correspondence with simple objects of the *Drinfel'd center* $\mathcal{Z}(\mathcal{C})$ labelling the superselection sectors of the original and dual model. Such a (simple) sector $Z \in \mathcal{I}_{\mathcal{Z}(\mathcal{C})}$ provides a certain (typically non-simple) boundary condition as well as a charge leaving that boundary condition invariant. In virtue of the Morita equivalence between $\mathcal{C}$ and $\mathcal{C}^\star_{\mathcal{M}}$, $\mathcal{Z}(\mathcal{C})$ is equivalent to $\mathcal{Z}(\mathcal{C}^\star_{\mathcal{M}})$ as a braided fusion category, and projectors on the topological sectors in both models are given by the minimal central idempotents of the $\mathfrak{T}_{\mathcal{R}|\mathcal{R}}$ and $\mathfrak{T}_{\mathcal{R}'|\mathcal{R}'}$ tube algebras. The duality $\mathcal{R} \to \mathcal{R}'$ then induces a permutation of the topological sectors, i.e. $\mathcal{Z}(\mathcal{C}) \xrightarrow{\sim} \mathcal{Z}(\mathcal{C}^\star_{\mathcal{M}})$, in accordance with the fact that not all boundary conditions in the dual model are compatible with a certain boundary condition in the original model, as is apparent from eq (40).

Let us now proceed by proposing a gauging map acting on the twisted Hilbert space eq. (39) which is unitarily equivalent to a (linear combination) of intertwining tubes eq. (40) [WBM$^+$14]. Requiring that the gauging map is constructed from a product of local commuting projectors acting on an initial trivial gauge field and that it acts diagonally on the boundary condition suggests following definition:

$$
\mathbb{G}^B_A := \sqrt{d_A}
$$

(41)

where, as above, every red strand is labelled by the algebra object $A$. The unitary circuit that converts $\mathbb{G}^B_A$ in a linear combination of tubes $\mathfrak{T}_{\mathcal{R}'|\mathcal{R}}$ parallels the one from the previous section. We begin by inflating all occurences of the comultiplication tensor in $\mathbb{G}^B_A$ using eq. (29). Subsequently we apply the gates defined in eq. (32). This results again in a collection of 'cups' which in turn can be removed by a depth 2 circuit, as discussed around eq. (34). As such, one ends up with

$$
\mathbb{G}^B_A \cong \sqrt{d_A}
$$

(42)

Finally, the boundary condition $B$ can be fused to the duality strands $X_A$ and $\bar{X}_A$ by sequentially

applying a variation of the gate eq. (32). The following linear combination of tubes is obtained:

$$\mathbb{G}_A^B \cong \sqrt{d_{X_A}} \sum_{\substack{X',B' \\ k,k'}} \quad \text{[diagram]} \quad . \tag{43}$$

The orientation reversing flag can then be absorbed in the fusion tensor to its right, at the cost of introducing extra quantum dimensions as per eq. (B.5). In the end, we obtain that

$$\mathbb{G}_A^B \cong \sum_{\substack{X',B' \\ k,k'}} \sqrt{\frac{d_{X_A} d_{X'}}{d_{B'}}} \times \mathfrak{T}_{\mathcal{R}'|\mathcal{R}}^{B,B',X,X',k,k'}. \tag{44}$$

SECTION 7

# Examples

*We illustrate our formalism by means of two examples corresponding to symmetries encoded in the fusion category of representations of $S_3$ and the Haagerup categories.*

### 7.1   Rep($S_3$)

As a first example, consider the fusion category $\mathsf{Rep}(S_3)$ of (finite-dimensional complex) representations of the permutation group $S_3$. An example Hamiltonian exhibiting $S_3$ symmetry is the spin-1/2 XXZ model. This model and all its dualities were considered in ref. [LDV24].

The simple objects of $\mathsf{Rep}(S_3)$ are denoted by $\mathcal{I}_{\mathsf{Rep}(S_3)} = \{\underline{1}, \underline{1}^*, \underline{2}\}$, and correspond to the trivial, sign and 2-dimensional irreducible representation respectively. Its non-trivial fusion rules are given by

$$\begin{aligned}
\underline{1}^* \otimes \underline{1}^* &\simeq \underline{1}, \\
\underline{1}^* \otimes \underline{2} &\simeq \underline{2} \oplus \underline{1}^* \simeq \underline{2}, \\
\underline{2} \otimes \underline{2} &\simeq \underline{1} \oplus \underline{1}^* \oplus \underline{2}.
\end{aligned} \tag{45}$$

Recall that the inequivalent indecomposable module categories of $\mathsf{Rep}(S_3)$ are in one-to-one correspondence with subgroups of $S_3$ up to conjugacy, of which there are four: $S_3$, $\mathbb{Z}_3$, $\mathbb{Z}_2$ and $\{1\}$. Given such a subgroup $H \subseteq S_3$ the corresponding module category is $\mathsf{Rep}(H)$ which is acted upon by $\mathsf{Rep}(S_3)$ via $\lambda \lhd \rho \simeq \lambda \otimes \mathrm{Res}_H^{S_3}(\rho)$, where $\mathrm{Res}_H^{S_3}$ denotes the restriction of $S_3$-representations to the subgroup $H$ and $\otimes$ stands for the usual tensor product of $H$-representations. For each of these module categories their Morita duals are computed to be

$$\begin{aligned}
\mathsf{Rep}(S_3)_{\mathsf{Rep}(S_3)}^\star &\simeq \mathsf{Rep}(S_3), \quad & \mathsf{Rep}(S_3)_{\mathsf{Rep}(\mathbb{Z}_3)}^\star &\simeq \mathsf{Vec}_{S_3}, \\
\mathsf{Rep}(S_3)_{\mathsf{Rep}(\mathbb{Z}_2)}^\star &\simeq \mathsf{Rep}(S_3), \quad & \mathsf{Rep}(S_3)_{\mathsf{Vec}}^\star &\simeq \mathsf{Vec}_{S_3}.
\end{aligned} \tag{46}$$

We now proceed to discuss the algebra objects in $\mathsf{Rep}(S_3)$ to which each of these module categories correspond.

Let us denote the irreps of $\mathbb{Z}_2$ by $\mathcal{I}_{\mathsf{Rep}(\mathbb{Z}_2)} = \{\underline{1}_{\mathbb{Z}_2}, \underline{1}_{\mathbb{Z}_2}^*\}$ where $\underline{1}_{\mathbb{Z}_2}^*$ stands for the sign representation of $\mathbb{Z}_2$. The restriction functor for this subgroup applied to the $S_3$-irreps yields $\mathrm{Res}_{\mathbb{Z}_2}^{S_3}(\underline{1}) \simeq \underline{1}_{\mathbb{Z}_2}$, $\mathrm{Res}_{\mathbb{Z}_2}^{S_3}(\underline{1}^*) \simeq \underline{1}_{\mathbb{Z}_2}^*$ and $\mathrm{Res}_{\mathbb{Z}_2}^{S_3}(\underline{2}) \simeq \underline{1}_{\mathbb{Z}_2} \oplus \underline{1}_{\mathbb{Z}_2}^*$. Making use of def. (A.15) and the $\mathsf{Rep}(\mathbb{Z}_2)$ fusion rules it is then straightforward to find that at the level of the objects, the internal Homs correspond to:

$$\begin{aligned}
\underline{\mathrm{Hom}}(\underline{1}_{\mathbb{Z}_2}, \underline{1}_{\mathbb{Z}_2}) &\simeq \underline{1} \oplus \underline{2}, \\
\underline{\mathrm{Hom}}(\underline{1}_{\mathbb{Z}_2}^*, \underline{1}_{\mathbb{Z}_2}^*) &\simeq \underline{1} \oplus \underline{2}.
\end{aligned} \tag{47}$$

All components of their respective multiplication morphisms evaluate to one except those given in terms of the F-symbols

$$\left(F^{1_{\mathbb{Z}_2}22}_{1_{\mathbb{Z}_2}}\right)^{1,11}_{1_{\mathbb{Z}_2},11} = \left(F^{1_{\mathbb{Z}_2}22}_{1_{\mathbb{Z}_2}}\right)^{1,11}_{1_{\mathbb{Z}_2},11} = \frac{1}{\sqrt{2}}$$

$$\left(F^{1_{\mathbb{Z}_2}22}_{1_{\mathbb{Z}_2}}\right)^{2,11}_{1_{\mathbb{Z}_2},11} = \frac{1}{\sqrt{2}}, \quad \left(F^{1^*_{\mathbb{Z}_2}22}_{1^*_{\mathbb{Z}_2}}\right)^{2,11}_{1^*_{\mathbb{Z}_2},11} = -\frac{1}{\sqrt{2}}. \tag{48}$$

These algebra structures are readily checked to be isomorphic. Equivalently, both inner Homs can be converted into each other by an automorphism of the module category given by the module functor $- \triangleleft 1^*$.

We write $\mathcal{I}_{\mathsf{Rep}(\mathbb{Z}_3)} = \{1_{\mathbb{Z}_3}, \omega_{\mathbb{Z}_3}, \omega^2_{\mathbb{Z}_3}\}$ for the three one-dimensional representations of $\mathbb{Z}_3$. Restrictions of the $S_3$-irreps to $\mathbb{Z}_3$ are computed to be $\mathrm{Res}^{S_3}_{\mathbb{Z}_3}(1) \simeq 1_{\mathbb{Z}_3}$, $\mathrm{Res}^{S_3}_{\mathbb{Z}_3}(1^*) \simeq 1_{\mathbb{Z}_3}$ and $\mathrm{Res}^{S_3}_{\mathbb{Z}_3}(2) \simeq \omega_{\mathbb{Z}_3} \oplus \omega^*_{\mathbb{Z}_3}$. The internal Homs are given by $\underline{\mathrm{Hom}}(\lambda, \lambda) \simeq 1 \oplus 1^*$ for all $\lambda \in \mathcal{I}_{\mathsf{Rep}(\mathbb{Z}_3)}$. The components of the multiplication map all evaluate to $\mu^{\lambda_1,\lambda_2}_{\lambda_1 \otimes \lambda_2} = \frac{1}{\sqrt{2}}$ for all $\lambda_1, \lambda_2 \in \{1, 1^*\}$.

Finally, considering $\mathsf{Vec}$ as module category, one finds as anticipated the regular representation of $S_3$ as corresponding algebra object:

$$\underline{\mathrm{Hom}}(\mathbb{C}, \mathbb{C}) \simeq 1 \oplus 1^* \oplus 2 \cdot 2. \tag{49}$$

Its algebra morphism is then specified by the F-symbol $F^{\mathbb{C}\lambda_1\lambda_2}_{\mathbb{C}}$ whose components evaluate to the (unitary) Clebsch-Gordan coefficients according to the prescription:

$$\left(F^{\mathbb{C}\lambda_1\lambda_2}_{\mathbb{C}}\right)^{\lambda_3,k1}_{\mathbb{C},ij} = \begin{bmatrix} \lambda_1 & \lambda_2 & | & \lambda_3 \\ i & j & | & k \end{bmatrix}, \tag{50}$$

where $\lambda_3 \in \lambda_1 \otimes \lambda_2$.

## 7.2 Haagerup categories

The Haagerup Morita class consists of three distinct unitary fusion categories denoted by $\mathcal{H}_1$, $\mathcal{H}_2$ and $\mathcal{H}_3$. The first two arise from the *Haagerup subfactor* proposed and constructed in refs. [Haa94, AH99], while in ref. [GS12] the existence of $\mathcal{H}_3$ was proven by providing an appropriate algebra object in $\mathcal{H}_1$ whose corresponding category of bimodules was hitherto unknown. The interest in the Haagerup subfactor stems from Jones' conjectured correspondence between subfactors and conformal field theories [Jon90]. Since the conjecture does not provide an explicit construction of the CFT, this has led to a plethora of recent studies in which both one-dimensional quantum models [HLO$^+$22, LDOV23, CdL24, BSN24, Jia24] as well as classical models [VLVD$^+$21] with $\mathcal{H}_3$ symmetry have been constructed and investigated using analytical and numerical methods, so as to shed light on the alleged Haagerup CFT.

The unitary fusion categories $\mathcal{H}_2$ and $\mathcal{H}_3$ are both rank six and have the same simple objects, denoted by $\mathcal{I}_{\mathcal{H}_2} = \mathcal{I}_{\mathcal{H}_3} = \{1, \alpha, \alpha^2, \rho, \alpha\rho, \alpha^2\rho\}$. They satisfy the same fusion rules, the non-trivial ones being given by

$$\alpha \otimes \alpha \simeq \alpha^2, \quad \alpha^i \otimes \rho \simeq \alpha^i\rho \simeq \rho \otimes \bar{\alpha}^i,$$
$$\alpha^2 \otimes \alpha \simeq 1, \quad \rho \otimes \rho \simeq 1 \oplus \rho \oplus \alpha\rho \oplus \alpha^2\rho. \tag{51}$$

The two categories differ in their unitary F-symbols, which can not be transmuted into each other via basis transformations on their multiplicity spaces. Both $\mathcal{H}_2$ and $\mathcal{H}_3$ possess a $\mathsf{Vec}_{\mathbb{Z}_3}$ subcategory consisting of the simple objects $\{1, \alpha, \alpha^2\}$.

The fusion category $\mathcal{H}_1$ on the other hand has four simple objects denoted by $\mathcal{I}_{\mathcal{H}_1} = \{1, \mu, \eta, \nu\}$. Its fusion rules are symmetric, and the non-trivial ones read

$$\mu \otimes \mu \simeq 1 \oplus \nu, \quad \nu \otimes \nu \simeq 1 \oplus 2\nu \oplus 2\eta \oplus \mu,$$
$$\mu \otimes \nu \simeq \mu \oplus \nu \oplus \eta, \quad \nu \otimes \eta \simeq 2\nu \oplus \eta \oplus \mu, \tag{52}$$
$$\mu \otimes \eta \simeq \nu \oplus \eta, \quad \eta \otimes \eta \simeq 1 \oplus \nu \oplus \eta \oplus \mu.$$

The F-symbols of $\mathcal{H}_1$, $\mathcal{H}_2$ and $\mathcal{H}_3$ were first calculated in refs. [BBW22, HL20, OSW19] respectively.

Since these three fusion categories are Morita equivalent, there exist invertible bimodule categories between them [GS12]. Let us denote these bimodule categories by $\mathcal{H}_{i,j}$, $i,j = 1, 2, 3$, such that $\left(\mathcal{H}_i\right)^\star_{\mathcal{H}_{i,j}} \simeq \mathcal{H}_j$, with the understanding that $\mathcal{H}_{j,i} \equiv \mathcal{H}_{i,j}^{\mathrm{op}}$ and $\mathcal{H}_{i,i} \equiv \mathcal{H}_i$.

The aforementioned quantum models with $\mathcal{H}_3$ symmetry can be obtained by choosing $\mathcal{M} = \mathcal{D} = \mathcal{H}_3$ as input to the models defined in eq. (18), and appropriate choices for the coefficients $\lambda_{i,n}$. Dual models with symmetries encoded in $\mathcal{H}_1$, $\mathcal{H}_2$ are then obtained by choosing the module categories $\mathcal{H}_{1,3}$, respectively $\mathcal{H}_{2,3}$, over the input category whose module associators are attached to ref. [BBW22]. Coincidentally, in virtue of the fact that $\mathsf{Fun}_{\mathcal{D}}(\mathcal{D}, \mathcal{M}) \simeq \mathcal{M}$, the category of duality operators is in those respective cases also given by $\mathcal{H}_{1,3}, \mathcal{H}_{2,3}$.

Let us first consider $\mathcal{H}_{1,3}$, whose simple objects we write as $\{\Gamma, \Gamma\alpha, \Gamma\alpha^2, \Lambda\}$. It follows from $N^\Gamma_{\Gamma,\alpha\rho} = N^\Gamma_{\Gamma,\alpha^2\rho} = 1$ that $\underline{\mathrm{Hom}}(\Gamma, \Gamma) \simeq \mathbb{1} \oplus \alpha\rho \oplus \alpha^2\rho$, as an object. The subset of F-symbols appearing in the algebra structure on $\underline{\mathrm{Hom}}(\Gamma, \Gamma)$ are given in ref. [BBW22]. In similar fashion one constructs the algebra objects $\underline{\mathrm{Hom}}(\Gamma\alpha, \Gamma\alpha)$, $\underline{\mathrm{Hom}}(\Gamma\alpha^2, \Gamma\alpha^2)$, which are related to $\underline{\mathrm{Hom}}(\Gamma, \Gamma)$ by conjugation with the invertible objects $\alpha^2$ and $\alpha$ of $\mathcal{H}_3$ respectively. Finally one constructs also the algebra object $\underline{\mathrm{Hom}}(\Lambda, \Lambda)$ which turns out to be equivalent to $\underline{\mathrm{Hom}}(\Lambda, \Lambda) \simeq \mathbb{1} \oplus \alpha\rho \oplus \alpha^2\rho$.

$\mathcal{H}_{2,3}$ has two simple objects $\mathcal{I}_{\mathcal{H}_{2,3}} \simeq \{\Omega, \Omega\rho\}$. Its fusion rules can be inferred from $\Omega \triangleleft \alpha^i \simeq \Omega$ and $\Omega \triangleleft \rho \simeq \Omega\rho$. From the first set of fusion rules it follows that the duality $\mathcal{H}_{2,3}$ amounts to gauging the $\mathsf{Vec}_{\mathbb{Z}_3}$ symmetry generated by $\alpha$, as indeed $\underline{\mathrm{Hom}}(\Omega, \Omega) \simeq \mathbb{1} \oplus \alpha \oplus \alpha^2$. The other module object gives rise to the Morita equivalent algebra object $\underline{\mathrm{Hom}}(\Omega\rho, \Omega\rho) \simeq \mathbb{1} \oplus \alpha \oplus \alpha^2 \oplus 3 \cdot (\rho \oplus \alpha\rho \oplus \alpha^2\rho)$. Likewise, one shows that $\mathcal{H}_3$ is obtained from gauging the $\mathsf{Vec}_{\mathbb{Z}_3}$ subsymmetry inside $\mathcal{H}_2$.

## Acknowledgements

We acknowledge Clement Delcamp, Jutho Haegeman, Gregor Schaumann, Sahand Seifnashri, Shu-Heng Shao and Boris De Vos for conversations at various stages of this work and feedback on the manuscript. This work has received funding from the Research Foundation Flanders (FWO) through Ph.D. fellowship No. 11O2423N awarded to BVDC, EOS (Grant No. 40007526), IBOF (Grant No. IBOF23/064), and BOF-GOA (Grant No. BOF23/GOA/021). It is also funded via UKRI grant EP/Z003342/1, and ERC-CoG SEQUAM (Grant Agreement No. 863476). JGR is funded by the FWF Erwin Schrödinger Program (Grant DOI 10.55776/J4796). DJW is supported by the Australian Research Council Discovery Early Career Research Award (DE220100625). L.L. is supported by an EPSRC Postdoctoral Fellowship (grant No. EP/Y020456/1).

# Categorical prerequisites

*In this section we introduce the necessary category theoretic tools, notations and conventions used throughout the manuscript. We refer to the standard references [EGNO15, FRS02] for a more elaborate exposition.*

## A.1 Fusion categories and algebra objects

Succinctly, a *unitary fusion category* (UFC) $\mathcal{C}$ consists of a collection of objects labeling topological defects whose set of (representatives of isomorphism classes of) simple objects is denoted $\mathcal{I}_{\mathcal{C}}$. The collection of defects is endowed with a monoidal product $\otimes : \mathcal{C} \times \mathcal{C} \to \mathcal{C}$ encoding their fusion. It is useful to introduce an integer-valued rank 3 tensor $N$ encoding the fusion rules as $X_1 \otimes X_2 \simeq \bigoplus_{X_3 \in \mathcal{I}_{\mathcal{C}}} N_{X_1 X_2}^{X_3} X_3$. Among the simple objects there exists a unique trivial defect $\mathbb{1}$ satisfying $\mathbb{1} \otimes X \simeq X \otimes \mathbb{1} \simeq X, \forall X \in \mathcal{I}_{\mathcal{C}}$. The fusion rules are associative up to an isomorphism which satisfies a consistency axiom known as the *pentagon equation*. Given an appropriate choice of basis for the junction spaces $\mathcal{C}(X_1 \otimes X_2, X_3)$, the components of this isomorphism can be expressed as unitary *F-symbols* defined graphically via:

$$\left( F_{X_4}^{X_1 X_2 X_3} \right)_{X_5, ij}^{X_6, kl} \qquad (A.1)$$

In the unitary gauge we additionally assume that

$$\left( F_X^{X \bar{X} X} \right)_{\mathbb{1}, 11}^{\mathbb{1}, 11} = \frac{\varkappa_X}{d_X} \qquad (A.2)$$

where $\varkappa_X = \pm 1$ denotes the *Frobenius-Schur indicator* of $X$, which satisfies $\varkappa_{\bar{X}} = \varkappa_X^*$, and the *quantum dimension* $d_X$ of the defect $X$ is the unique solution to

$$d_X > 0,$$
$$d_{X_1} d_{X_2} = \sum_{X_3 \in \mathcal{I}_{\mathcal{C}}} N_{X_1 X_2}^{X_3} d_{X_3}, \qquad (A.3)$$

for all $X, X_1, X_2 \in \mathcal{I}_{\mathcal{C}}$.

An *algebra object*, henceforth often called simply *algebra*, in a UFC $\mathcal{C}$ is a triple $(A, \mu, \eta)$ consisting of an object $A \simeq \bigoplus_{X \in \mathcal{I}_{\mathcal{C}}} N_X^A X$, with $N_X^A = \dim_{\mathbb{C}} \mathcal{C}(X, A)$, together with a *multiplication morphism* $\mu \in \mathcal{C}(A \otimes A, A)$ that satisfies following associativity condition on the nose

$$(A.4)$$

and *unit map* $\eta \in \mathcal{C}(\mathbb{1}, A)$, denoted in our graphical calculus by a white node, satisfying

$$= \mathrm{id}_A. \qquad (A.5)$$

It is a useful fact that $N_X^A \leqslant \lfloor d_X \rfloor$ for the algebra objects considered in this manuscript. We also assume that all algebra objects are *haploid*, meaning that $A$ contains the monoidal unit $\mathbb{1}$ of $\mathcal{C}$ with multiplicity 1, or thus $N_{\mathbb{1}}^A = 1$. Indeed, by virtue of [Ost01, Thm. 3.1] every Morita class of (semisimple) indecomposable algebra objects contains at least one haploid representative. An algebra object $(A, \mu, \eta)$ is called *symmetric* if there exists a morphism $\varepsilon \in \mathcal{C}(A, \mathbb{1})$ so that

$$
\tag{A.6}
$$

where the flags depict respectively the right and left coevalution morphisms of $A$. It can be shown [FRS02, Cor. 3.10] that Haploid algebras are symmetric for all $\varepsilon \in \mathcal{C}(A, \mathbb{1})$ in virtue of their defining property $\dim_{\mathbb{C}} \mathcal{C}(\mathbb{1}, A) = 1$.

One can then define an analogous *coalgebra object* as a triple $(A, \Delta, \varepsilon)$ where $A$ is an object in $\mathcal{C}$, $\Delta \in \mathcal{C}(A, A \otimes A)$ is a *comultiplication map* satisfying a coassociativity condition akin to eq. (A.4) and $\varepsilon \in \mathcal{C}(A, \mathbb{1})$ is a *counit map* which together with $\Delta$ is required to obey a mirrored version of eq. (A.5). A *special* algebra object is an object that is both an algebra $(A, \mu, \eta)$ and coalgebra $(A, \Delta, \varepsilon)$ for which

$$
A \left\langle \begin{matrix} \Delta \\ \mu \end{matrix} \right\rangle A \;=\; \beta_A \cdot \mathrm{id}_A, \qquad \text{and} \qquad A \Big| \;=\; \beta_{\mathbb{1}} \cdot \mathrm{id}_{\mathbb{1}}, \qquad \beta_A \cdot \beta_{\mathbb{1}} = d_A \tag{A.7}
$$

with the quantum dimension of the algebra object $A$ given by $d_A = \sum_{X \in \mathcal{I}_\mathcal{C}} N_X^A d_X$. To reduce the occurrence of quantum dimensions in the main text, we adopt without loss of generality the convention that $\beta_A = 1$, $\beta_{\mathbb{1}} = d_A$.

Proceeding as such, a *Frobenius algebra object* is a quintuple $(A, \mu, \eta, \Delta, \varepsilon)$ where $(A, \mu, \eta)$ constitutes an algebra and $(A, \Delta, \varepsilon)$ a coalgebra, such that the multiplication and comultiplication map jointly obey the *Frobenius relations*

$$
\tag{A.8}
$$

Considering an algebra $(A, \Delta, u)$ in $\mathcal{C}$, the multiplication morphism can be written in components by means of

$$
= \sum_l \mu_{(X_3,k),l}^{(X_1,i),(X_2,j)} \tag{A.9}
$$

where $i, j, k$ and $l$ label orthonormal basis vectors in the vector spaces $\mathcal{C}(X_1, A)$, $\mathcal{C}(X_2, A)$, $\mathcal{C}(A, X_3)$ and $\mathcal{C}(X_1 \otimes X_2, X_3)$ respectively. Analogously one defines the tensor $\Delta_{(X_1,i),(X_2,j)}^{(X_3,k),l}$ for the comultiplication map.

Two algebra objects are *isomorphic* when they are related by unitary basis transformations on the multiplicity spaces $\mathcal{C}(X_i, A)$ and $\mathcal{C}(A, X_i)$.

## A.2 Module categories and module functors

Given a UFC $\mathcal{D}$, a (unitary finite indecomposable semisimple) *right module category* $\mathcal{R}$ over $\mathcal{D}$ is a semisimple category whose set of (representative objects of isomorphism classes of) simple objects is written as $\mathcal{I}_{\mathcal{R}}$ and which is equipped with a right action bifunctor $\triangleleft : \mathcal{R} \times \mathcal{D} \to \mathcal{R}$. Up to an isomorphism the action is associative. In an appropriate basis of the junction spaces, this isomorphism can once more be expressed as following unitary F-symbols:

$$\text{(A.10)}$$

These F-symbols satisfy a mixed pentagon equation which also involves the F-symbols of $\mathcal{D}$. We choose a basis for the Hom spaces such that

$$\left(F_{R_2}^{R_1 \mathbb{1} X_2}\right)_{R_1, 1i}^{X_2, 1j} = \left(F_{R_2}^{R_1 X_1 \mathbb{1}}\right)_{R_2, i1}^{X_1, 1j} = \delta_{i,j}. \tag{A.11}$$

Also for the module objects $R \in \mathcal{I}_{\mathcal{R}}$ we define quantum dimensions, which are uniquely specified by [ENOM09]:

$$
\begin{aligned}
d_R &> 0, \\
d_{R_1} d_X &= \sum_{R_2 \in \mathcal{I}_{\mathcal{R}}} N_{R_1 X}^{R_2} d_{R_2}, \\
\sum_{R \in \mathcal{I}_{\mathcal{R}}} d_R^2 &= \sum_{X \in \mathcal{I}_{\mathcal{C}}} d_X^2,
\end{aligned}
\tag{A.12}
$$

for all $R, R_1 \in \mathcal{I}_{\mathcal{R}}, X \in \mathcal{I}_{\mathcal{C}}$. Analogously, one defines a *left* module category over $\mathcal{D}$ by mirroring the above definition.

Crucial to the construction of the duality intertwiners and symmetry MPOs is the notion of $\mathcal{D}$-*module functors* between (right) $\mathcal{D}$-module categories $\mathcal{R}$ and $\mathcal{R}'$. Concretely, a module functor between $\mathcal{R}$ and $\mathcal{R}'$ is a functor $\mathfrak{F} : \mathcal{R} \to \mathcal{R}'$, which respects the module structures of $\mathcal{R}$ and $\mathcal{R}'$ in the sense that there exists an isomorphism $\omega$ whose components are given by $\omega^{RY} : \mathfrak{F}(R \triangleleft Y) \xrightarrow{\sim} \mathfrak{F}(R) \triangleleft Y$ for all $R, Y$ in $\mathcal{I}_{\mathcal{R}}$ and $\mathcal{I}_{\mathcal{D}}$ respectively and where $\triangleleft$ and $\triangleleft$ denote the action functors of $\mathcal{R}$ and $\mathcal{R}'$ respectively. Such module functors are organized in a (semisimple) category $\mathsf{Fun}_{\mathcal{D}}(\mathcal{R}, \mathcal{R}')$ [EGNO15]. Given a simple object $X$ in $\mathsf{Fun}_{\mathcal{D}}(\mathcal{R}, \mathcal{R}')$, let us write $^X\omega$ for its corresponding isomorphism. Its components can be expressed via the diagram

$$\text{(A.13)}$$

Consider now three right module categories $\mathcal{M}, \mathcal{N}, \mathcal{O}$ over $\mathcal{D}$. Module functors in $\mathsf{Fun}_{\mathcal{D}}(\mathcal{M}, \mathcal{N})$ and $\mathsf{Fun}_{\mathcal{D}}(\mathcal{N}, \mathcal{O})$ can be composed, resulting in a module functor in $\mathsf{Fun}_{\mathcal{D}}(\mathcal{M}, \mathcal{O})$. Being semisimple, we can consider simple module functors $X_1, X_2$ in $\mathcal{I}_{\mathsf{Fun}_{\mathcal{D}}(\mathcal{N}, \mathcal{O})}$ and $\mathcal{I}_{\mathsf{Fun}_{\mathcal{D}}(\mathcal{M}, \mathcal{N})}$ respectively. Decomposing the composition of $X_1$ and $X_2$ in simples in $\mathcal{I}_{\mathsf{Fun}_{\mathcal{D}}(\mathcal{M}, \mathcal{O})}$ then happens via unitary complex matrices

whose entries are defined in terms of following string diagrams:

$$
\begin{array}{c}
\includegraphics{A14left}
\end{array}
= \sum_{N,kl} (F_O^{X_1 X_2 M})_{X_3,ij}^{N,kl} \quad
\begin{array}{c}
\includegraphics{A14right}
\end{array} , \tag{A.14}
$$

in which $i,j,k,l$ are indices in the appropriate junction spaces.

### A.3 Internal Hom objects and algebras

Given a right module category $\mathcal{M}$ over $\mathcal{C}$, an *internal Hom object* [Ost01] for any two objects $M_1, M_2$ is an object $\underline{\mathrm{Hom}}(M_1, M_2)$ in $\mathcal{C}$ together with a collection of isomorphisms

$$
\mathcal{M}(M_1 \lhd X, M_2) \simeq \mathcal{C}(X, \underline{\mathrm{Hom}}(M_1, M_2)), \tag{A.15}
$$

satisfying a number of naturality conditions specified in ref. [Sch13]. Crucially, for any simple object $M \in \mathcal{I}_{\mathcal{M}}$, $\underline{\mathrm{Hom}}(M, M)$ is guaranteed to be a special haploid symmetric Frobenius algebra object, as demonstrated in ref. [Sch13]. Let us spell this out explicitly. Notating $A \equiv \underline{\mathrm{Hom}}(M, M)$, it readily follows from (A.15) that the decomposition of $A$ in simple objects $\mathcal{C}$ is given by $A \simeq \bigoplus_{X \in \mathcal{I}_{\mathcal{C}}} N_{MX}^M X$, where $N_{MX}^M \equiv \dim_{\mathbb{C}} \mathcal{M}(M \lhd X, M)$. The components of the multiplication and comultiplication maps can be related to the right module associator (A.10) via

$$
\mu_{(X_3,k),l}^{(X_1,i),(X_2,j)} = \frac{1}{d_M} \sqrt{\frac{d_{X_1} d_{X_2}}{d_{X_3}}} (F_M^{M X_1 X_2})_{M,ij}^{X_3,lk},
$$

$$
\Delta_{(X_1,i),(X_2,j)}^{(X_3,k),l} = \frac{1}{d_M} \sqrt{\frac{d_{X_1} d_{X_2}}{d_{X_3}}} (\bar{F}_M^{M X_1 X_2})_{M,ij}^{X_3,lk}, \tag{A.16}
$$

in which once more the isomorphisms (A.15) were leveraged. In this normalisation of the multiplication and comultiplication map and the F-symbol eq. (A.11), the unique component of the unit and counit labelled by $\eta_1$ and $\varepsilon_1$ respectively is given by

$$
\eta_1 = \varepsilon_1 = d_M, \tag{A.17}
$$

which is verified to be compatible with the normalization in eq. (A.7).

SECTION B

## Orientation-reversing flags

Let $X$ be a simple object in $\mathsf{Fun}_{\mathcal{D}}(\mathcal{R}, \mathcal{R}')$ in what follows. Orientation-reversing 'flags' for $X$ are defined in terms of the fusion (splitting) tensors by terminating them in the vacuum label $\mathbb{1}$:

$$
\begin{array}{c}
\includegraphics{B1top}
\end{array}
\equiv \sum_{i,j}
\begin{array}{c}
\includegraphics{B1mid}
\end{array}
|RXR', i\rangle \otimes |R'\bar{X}R, j\rangle
$$

$$
\equiv \sum_{i,j} \sqrt{d_X} \times
\begin{array}{c}
\includegraphics{B1bot}
\end{array}
|RXR', i\rangle \otimes |R'\bar{X}R, j\rangle , \tag{B.1}
$$

and

$$\sum_{i,j} \left[ \cdots \right] \langle RXR', i| \otimes \langle R'\bar{X}R, j| \tag{B.2}$$

$$\equiv \sum_{i,j} \sqrt{d_X} \times \left[ \cdots \right] \langle RXR', i| \otimes \langle R'\bar{X}R, j| .$$

In these definitions, $\mathbb{1}$ denotes the monoidal unit in the fusion category $\mathcal{C}_{\mathcal{M}}^{\star}$, and 1 stands for the sole basis vector in $\mathcal{R}'(\mathbb{1} \triangleright R, R) \simeq \mathbb{C}$.

The Frobenius–Schur indicator, which for module objects is defined in a way analogous to eq. (A.2), can be used to reverse the orientation of the flags, concretely:

$$\left[ \cdots \right] = \varkappa_X^* \left[ \cdots \right] , \tag{B.3}$$

$$\left[ \cdots \right] = \varkappa_X \left[ \cdots \right] .$$

The flags satisfy a number of properties generalising those spelled out in refs. [BMW+15, LFH+20]. Most notably, they satisfy a collection of 'snake axioms' stating that oppositely paired flags cancel in the sense that eg.

$$\left[ \cdots \right] = \left[ \cdots \right] . \tag{B.4}$$

As such, these flags implement the evaluation and coevaluation maps of $X$, $\mathrm{ev}_X : \bar{X} \otimes X \to \mathbb{1}$, respectively $\mathrm{coev}_X : \mathbb{1} \to X \otimes \bar{X}$, in terms of triple line tensors.

Moreover, these flags can be used to bend lines on the fusion and splitting tensors as follows:

$$\left[ \cdots \right] = \sum_j \underbrace{\sqrt{d_{X_1}} \big( \bar{F}_{X_2}^{\bar{X}_1 X_1 X_2} \big)_{\mathbb{1},11}^{X_3,ij}}_{\sqrt{\frac{d_{X_3}}{d_{X_2}}} \big( \bar{A}_{X_3}^{X_1 X_2} \big)_{ij}} \left[ \cdots \right] , \tag{B.5}$$

$$\left[ \cdots \right] = \sum_j \underbrace{\sqrt{d_{X_2}} \big( F_{X_1}^{X_1 X_2 \bar{X}_2} \big)_{X_3,ij}^{\mathbb{1},11}}_{\sqrt{\frac{d_{X_3}}{d_{X_1}}} \big( B_{X_3}^{X_1 X_2} \big)_{ij}} \left[ \cdots \right] .$$

The unitary matrices $A$ and $B$ are defined as [Kit05]

$$\left(A_{X_3}^{X_1X_2}\right)_{ij} = \sqrt{\frac{d_{X_1}d_{X_2}}{d_{X_3}}}\left(F_{X_2}^{\bar{X}_1X_1X_2}\right)_{\mathbb{1},11}^{X_3,ij},$$

$$\left(B_{X_3}^{X_1X_2}\right)_{ij} = \sqrt{\frac{d_{X_1}d_{X_2}}{d_{X_3}}}\left(F_{X_1}^{X_1X_2\bar{X}_2}\right)_{X_3,ij}^{\mathbb{1},11}.$$

(B.6)

SECTION C

# Proof (co)algebra multiplication

Here, we showcase that the inflated comultiplication tensor (29) satisfies the comultiplication axiom by making use of our graphical calculus [FRS03, Sch13]. Invoking the orientation-reversing flags from section B one finds:

(C.1)

$$= \sum_{\substack{\{X_j\}_j, \\ \{i_j\}_j}} \frac{\sqrt{d_{X_1} d_{X_2} d_{X_3} d_{X_4}} \, d_{X_6}}{d_{X_A}^4}$$ 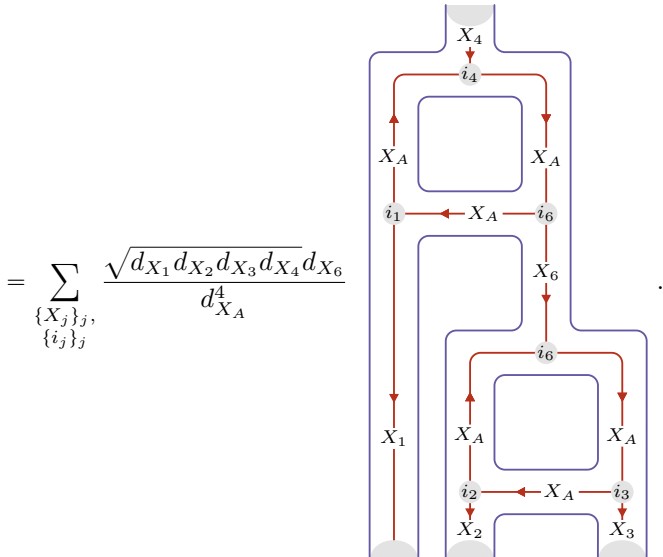 .

## – References –

[AFM20]     D. Aasen, P. Fendley, and R. S. K. Mong, *Topological Defects on the Lattice: Dualities and Degeneracies*, `arXiv:2008.08598 [cond-mat.stat-mech]`.

[AH99]     M. Asaeda and U. Haagerup, *Exotic subfactors of finite depth with Jones indices* $(5 + \sqrt{13})/2$ *and* $(5 + \sqrt{17})/2$, Communications in mathematical physics **202** (1999) 1–63.

[AMF16]     D. Aasen, R. S. K. Mong, and P. Fendley, *Topological Defects on the Lattice I: The Ising model*, J. Phys. A **49** (2016) 354001, `arXiv:1601.07185 [cond-mat.stat-mech]`.

[BAV09]     O. Buerschaper, M. Aguado, and G. Vidal, *Explicit tensor network representation for the ground states of string-net models*, Phys. Rev. B **79** (2009) 085119.

[BBW22]     D. Barter, J. C. Bridgeman, and R. Wolf, *Computing associators of endomorphism fusion categories*, SciPost Phys. **13** (2022) 029.

[Ben14]     Y. BenTov, *Fermion masses without symmetry breaking in two spacetime dimensions*, JHEP **07** (2015) 034, `arXiv:1412.0154 [cond-mat.str-el]`.

[BG17]     M. Buican and A. Gromov, *Anyonic Chains, Topological Defects, and Conformal Field Theory*, Commun. Math. Phys. **356** (2017) 1017–1056, `arXiv:1701.02800 [hep-th]`.

[BLV22]     J. C. Bridgeman, L. Lootens, and F. Verstraete, *Invertible Bimodule Categories and Generalized Schur Orthogonality*, Commun. Math. Phys. **402** (2023) 2691–2714, `arXiv:2211.01947 [math.QA]`.

[BMW+15]     N. Bultinck, M. Mariën, D. J. Williamson, M. B. Şahinoğlu, J. Haegeman, and F. Verstraete, *Anyons and matrix product operator algebras*, Annals Phys. **378** (2017) 183–233, `arXiv:1511.08090 [cond-mat.str-el]`.

[BSN24]     L. E. Bottini and S. Schafer-Nameki, *A Gapless Phase with Haagerup Symmetry*, `arXiv:2410.19040 [hep-th]`.

[BT17]     L. Bhardwaj and Y. Tachikawa, *On finite symmetries and their gauging in two dimensions*, JHEP **03** (2018) 189, `arXiv:1704.02330 [hep-th]`.

[CAW24]     A. Chatterjee, O. M. Aksoy, and X.-G. Wen, *Quantum phases and transitions in spin chains with non-invertible symmetries*, SciPost Phys. **17** (2024) 115, `arXiv:2405.05331 [cond-mat.str-el]`.

[CdL24]     L. Corcoran and M. de Leeuw, *Integrable and critical Haagerup spin chains*, `arXiv:2410.16356 [cond-mat.stat-mech]`.

[CLW11]     X. Chen, Z.-X. Liu, and X.-G. Wen, *Two-dimensional symmetry-protected topological orders and their protected gapless edge excitations*, Phys. Rev. B **84** (2011) 235141, `arXiv:1106.4752 [cond-mat.str-el]`.

[CON11]     E. Cobanera, G. Ortiz, and Z. Nussinov, *The Bond-Algebraic Approach to Dualities*, Adv. Phys. **60** (2011) 679–798, `arXiv:1103.2776 [cond-mat.stat-mech]`.

[CR12]     N. Carqueville and I. Runkel, *Orbifold completion of defect bicategories*, Quantum Topol. **7** (2016) 203–279, `arXiv:1210.6363 [math.QA]`.

[CV14]     X. Chen and A. Vishwanath, *Towards Gauging Time-Reversal Symmetry: A Tensor Network Approach*, Phys. Rev. X **5** (2015) 041034, `arXiv:1401.3736 [cond-mat.str-el]`.

[cWB+14]   M. B. Şahinoğlu, D. Williamson, N. Bultinck, M. Mariën, J. Haegeman, N. Schuch, and F. Verstraete, *Characterizing Topological Order with Matrix Product Operators*, Annales Henri Poincare **22** (2021) 563–592, arXiv:1409.2150 [quant-ph].

[DLWW23]   O. Diatlyk, C. Luo, Y. Wang, and Q. Weller, *Gauging non-invertible symmetries: topological interfaces and generalized orbifold groupoid in 2d QFT*, JHEP **03** (2024) 127, arXiv:2311.17044 [hep-th].

[DT23]   C. Delcamp and A. Tiwari, *Higher categorical symmetries and gauging in two-dimensional spin systems*, SciPost Phys. **16** (2024) 110, arXiv:2301.01259 [hep-th].

[EGNO15]   P. Etingof, S. Gelaki, D. Nikshych, and V. Ostrik, *Tensor categories*, American Mathematical Society, 2015.

[ENOM09]   P. Etingof, D. Nikshych, V. Ostrik, and w. a. a. b. E. Meir, *Fusion categories and homotopy theory*, arXiv:0909.3140 [math.QA].

[FMT22]   D. S. Freed, G. W. Moore, and C. Teleman, *Topological symmetry in quantum field theory*, arXiv:2209.07471 [hep-th].

[FRBC25]   A. Franco-Rubio, A. Bochniak, and J. I. Cirac, *Symmetry defects and gauging for quantum states with matrix product unitary symmetries*, arXiv:2502.20257 [quant-ph].

[FRS02]   J. Fuchs, I. Runkel, and C. Schweigert, *TFT construction of RCFT correlators 1. Partition functions*, Nucl. Phys. B **646** (2002) 353–497, arXiv:hep-th/0204148.

[FRS03]   J. Fuchs, I. Runkel, and C. Schweigert, *TFT construction of RCFT correlators. 2. Unoriented world sheets*, Nucl. Phys. B **678** (2004) 511–637, arXiv:hep-th/0306164.

[FRS04]   J. Fuchs, I. Runkel, and C. Schweigert, *TFT construction of RCFT correlators. 3. Simple currents*, Nucl. Phys. B **694** (2004) 277–353, arXiv:hep-th/0403157.

[FTL+06]   A. Feiguin, S. Trebst, A. W. W. Ludwig, M. Troyer, A. Kitaev, Z. Wang, and M. H. Freedman, *Interacting anyons in topological quantum liquids: The golden chain*, Phys. Rev. Lett. **98** (2007) 160409, arXiv:cond-mat/0612341.

[GKSW14]   D. Gaiotto, A. Kapustin, N. Seiberg, and B. Willett, *Generalized Global Symmetries*, JHEP **02** (2015) 172, arXiv:1412.5148 [hep-th].

[GRK22]   J. Garre-Rubio and I. Kull, *Gauging quantum states with nonanomalous matrix product operator symmetries*, Phys. Rev. B **107** (2023) 075137, arXiv:2209.07355 [quant-ph].

[GRLM22]   J. Garre-Rubio, L. Lootens, and A. Molnár, *Classifying phases protected by matrix product operator symmetries using matrix product states*, Quantum **7** (2023) 927, arXiv:2203.12563 [cond-mat.str-el].

[GS12]   P. Grossman and N. Snyder, *Quantum subgroups of the haagerup fusion categories*, Communications in Mathematical Physics **311** (2012) 617–643.

[Haa94]   U. Haagerup, *Principal graphs of subfactors in the index range $4 < [m : n] < 3 + \sqrt{2}$*, Subfactors (1994) .

[HEH+15]   M. Hauru, G. Evenbly, W. W. Ho, D. Gaiotto, and G. Vidal, *Topological conformal defects with tensor networks*, Phys. Rev. B **94** (2016) 115125, arXiv:1512.03846 [cond-mat.str-el].

[HL20]   T.-C. Huang and Y.-H. Lin, *The F-Symbols for Transparent Haagerup-Izumi Categories with $G = \mathbb{Z}_{2n+1}$*, arXiv:2007.00670 [math.CT].

[HLO⁺22]    T.-C. Huang, Y.-H. Lin, K. Ohmori, Y. Tachikawa, and M. Tezuka, *Numerical evidence for a haagerup conformal field theory*, Phys. Rev. Lett. **128** (2022) 231603.

[HV16]      J. Haegeman and F. Verstraete, *Diagonalizing Transfer Matrices and Matrix Product Operators: A Medley of Exact and Computational Methods*, Ann. Rev. Condensed Matter Phys. **8** (2017) 355–406, `arXiv:1611.08519 [cond-mat.str-el]`.

[HVAS⁺15]   J. Haegeman, K. Van Acoleyen, N. Schuch, J. I. Cirac, and F. Verstraete, *Gauging quantum states: From global to local symmetries in many-body systems*, Phys. Rev. X **5** (2015) 011024.

[Jia24]     Z. Jia, *Quantum Cluster State Model with Haagerup Fusion Category Symmetry*, `arXiv:2412.19657 [math.QA]`.

[Jon90]     V. F. Jones, *von Neumann algebras in mathematics and physics*, American Mathematical Society, 1990.

[Jon23]     C. Jones, *DHR bimodules of quasi-local algebras and symmetric quantum cellular automata*, `arXiv:2304.00068 [math-ph]`.

[JR15]      S. Jiang and Y. Ran, *Symmetric tensor networks and practical simulation algorithms to sharply identify classes of quantum phases distinguishable by short-range physics*, Phys. Rev. B **92** (2015) 104414, `arXiv:1505.03171 [cond-mat.str-el]`.

[JSW24]     C. Jones, K. Schatz, and D. J. Williamson, *Quantum cellular automata and categorical dualities of spin chains*, `arXiv:2410.08884 [math-ph]`.

[JW28]      P. Jordan and E. P. Wigner, *About the Pauli exclusion principle*, Z. Phys. **47** (1928) 631–651.

[Kit05]     A. Kitaev, *Anyons in an exactly solved model and beyond*, Annals Phys. **321** (2006) 2–111, `arXiv:cond-mat/0506438`.

[KORS20]    Z. Komargodski, K. Ohmori, K. Roumpedakis, and S. Seifnashri, *Symmetries and strings of adjoint QCD$_2$*, JHEP **03** (2021) 103, `arXiv:2008.07567 [hep-th]`.

[KT92]      T. Kennedy and H. Tasaki, *Hidden Z2×Z2 symmetry breaking in Haldane-gap antiferromagnets*, Phys. Rev. B **45** (1992) 304.

[KW41]      H. A. Kramers and G. H. Wannier, *Statistics of the two-dimensional ferromagnet. Part 1.*, Phys. Rev. **60** (1941) 252–262.

[LDOV23]    L. Lootens, C. Delcamp, G. Ortiz, and F. Verstraete, *Dualities in one-dimensional quantum lattice models: Symmetric hamiltonians and matrix product operator intertwiners*, PRX Quantum **4** (2023) 020357.

[LDV24]     L. Lootens, C. Delcamp, and F. Verstraete, *Dualities in one-dimensional quantum lattice models: Topological sectors*, PRX Quantum **5** (2024) 010338.

[LDV25]     L. Lootens, C. Delcamp, and F. Verstraete, *Entanglement and the density matrix renormalization group in the generalized landau paradigm*, Nature Physics (2025) 1–7.

[LDWV23]    L. Lootens, C. Delcamp, D. Williamson, and F. Verstraete, *Low-depth unitary quantum circuits for dualities in one-dimensional quantum lattice models*, `arXiv:2311.01439 [quant-ph]`.

[LFH⁺20]    L. Lootens, J. Fuchs, J. Haegeman, C. Schweigert, and F. Verstraete, *Matrix product operator symmetries and intertwiners in string-nets with domain walls*, SciPost Phys. **10** (2021) 053, `arXiv:2008.11187 [quant-ph]`.

[LOZ23]     L. Li, M. Oshikawa, and Y. Zheng, *Noninvertible duality transformation between symmetry-protected topological and spontaneous symmetry breaking phases*, Phys. Rev. B **108** (2023) 214429, `arXiv:2301.07899 [cond-mat.str-el]`.

[LSY24]     D.-C. Lu, Z. Sun, and Y.-Z. You, *Realizing triality and p-ality by lattice twisted gauging in (1+1)d quantum spin systems*, SciPost Phys. **17** (2024) 136, `arXiv:2405.14939 [cond-mat.str-el]`.

[LVDCSV21]  L. Lootens, B. Vancraeynest-De Cuiper, N. Schuch, and F. Verstraete, *Mapping between Morita-equivalent string-net states with a constant depth quantum circuit*, Phys. Rev. B **105** (2022) 085130, `arXiv:2112.12757 [quant-ph]`.

[MABT23]    H. Moradi, O. M. Aksoy, J. H. Bardarson, and A. Tiwari, *Symmetry fractionalization, mixed-anomalies and dualities in quantum spin models with generalized symmetries*, `arXiv:2307.01266 [cond-mat.str-el]`.

[MMT22]     H. Moradi, S. F. Moosavian, and A. Tiwari, *Topological holography: Towards a unification of Landau and beyond-Landau physics*, SciPost Phys. Core **6** (2023) 066, `arXiv:2207.10712 [cond-mat.str-el]`.

[NY18]      S. Neshveyev and M. Yamashita, *A few remarks on the tube algebra of a monoidal category*, Proceedings of the Edinburgh Mathematical Society **61** (2018) 735–758.

[Osh92]     M. Oshikawa, *Hidden $\mathbb{Z}_2 \times \mathbb{Z}_2$ symmetry in quantum spin chains with arbitrary integer spin*, Journal of Physics: Condensed Matter **4** (1992) 7469.

[Ost01]     V. Ostrik, *Module categories, weak Hopf algebras and modular invariants*, Transform. Groups **8** (2003) 177–206, `arXiv:math/0111139`.

[OSW19]     T. J. Osborne, D. E. Stiegemann, and R. Wolf, *The F-Symbols for the H3 Fusion Category*, `arXiv:1906.01322 [math.CT]`.

[PZ00]      V. B. Petkova and J. B. Zuber, *Generalized twisted partition functions*, Phys. Lett. B **504** (2001) 157–164, `arXiv:hep-th/0011021`.

[Rad18]     D. Radicevic, *Spin Structures and Exact Dualities in Low Dimensions*, `arXiv:1809.07757 [hep-th]`.

[Sch13]     G. Schaumann, *Traces on module categories over fusion categories*, Journal of Algebra **379** (2013) 382–425.

[Sha23]     S.-H. Shao, *What's Done Cannot Be Undone: TASI Lectures on Non-Invertible Symmetries*, `arXiv:2308.00747 [hep-th]`.

[SN23]      S. Schafer-Nameki, *ICTP lectures on (non-)invertible generalized symmetries*, Phys. Rept. **1063** (2024) 1–55, `arXiv:2305.18296 [hep-th]`.

[SS24]      S. Seifnashri and S.-H. Shao, *Cluster State as a Noninvertible Symmetry-Protected Topological Phase*, Phys. Rev. Lett. **133** (2024) 116601, `arXiv:2404.01369 [cond-mat.str-el]`.

[SSS24]     N. Seiberg, S. Seifnashri, and S.-H. Shao, *Non-invertible symmetries and LSM-type constraints on a tensor product Hilbert space*, SciPost Phys. **16** (2024) 154, `arXiv:2401.12281 [cond-mat.str-el]`.

[SSY25]     S. Seifnashri, S.-H. Shao, and X. Yang, *Gauging non-invertible symmetries on the lattice*, SciPost Phys. **19** (2025) 063, `arXiv:2503.02925 [cond-mat.str-el]`.

[Tac17]     Y. Tachikawa, *On gauging finite subgroups*, SciPost Phys. **8** (2020) 015, arXiv:1712.09542 [hep-th].

[TW19]      R. Thorngren and Y. Wang, *Fusion category symmetry. Part I. Anomaly in-flow and gapped phases*, JHEP **04** (2024) 132, arXiv:1912.02817 [hep-th].

[VBW+18]    R. Vanhove, M. Bal, D. J. Williamson, N. Bultinck, J. Haegeman, and F. Verstraete, *Mapping topological to conformal field theories through strange correlators*, Phys. Rev. Lett. **121** (2018) 177203, arXiv:1801.05959 [quant-ph].

[VDCBD+22]  B. Vancraeynest-De Cuiper, J. C. Bridgeman, N. Dewolf, J. Haegeman, and F. Verstraete, *One-dimensional symmetric phases protected by frieze symmetries*, Phys. Rev. B **107** (2023) 115123, arXiv:2202.12880 [cond-mat.str-el].

[VDCD25]    B. Vancraeynest-De Cuiper and C. Delcamp, *Twisted gauging and topological sectors in (2+1)d Abelian lattice gauge theories*, SciPost Phys. **19** (2025) 054, arXiv:2501.16301 [cond-mat.str-el].

[VDCGR24]   B. Vancraeynest-De Cuiper and J. Garre-Rubio, *Systematic construction of stabilizer codes via gauging abelian boundary symmetries*, Quantum **9** (2025) 1852, arXiv:2410.09044 [quant-ph].

[VLTV21]    R. Vanhove, L. Lootens, H.-H. Tu, and F. Verstraete, *Topological aspects of the critical three-state Potts model*, J. Phys. A **55** (2022) 235002, arXiv:2107.11177 [math-ph].

[VLVD+21]   R. Vanhove, L. Lootens, M. Van Damme, R. Wolf, T. J. Osborne, J. Haegeman, and F. Verstraete, *Critical Lattice Model for a Haagerup Conformal Field Theory*, Phys. Rev. Lett. **128** (2022) 231602, arXiv:2110.03532 [cond-mat.stat-mech].

[VWPGC06]   F. Verstraete, M. M. Wolf, D. Perez-Garcia, and J. I. Cirac, *Criticality, the area law, and the computational power of projected entangled pair states*, Phys. Rev. Lett. **96** (2006) 220601.

[WBM+14]    D. J. Williamson, N. Bultinck, M. Mariën, M. B. Şahinoğlu, J. Haegeman, and F. Verstraete, *Matrix product operators for symmetry-protected topological phases: Gauging and edge theories*, Phys. Rev. B **94** (2016) 205150, arXiv:1412.5604 [quant-ph].

[WBV17]     D. J. Williamson, N. Bultinck, and F. Verstraete, *Symmetry-enriched topological order in tensor networks: Defects, gauging and anyon condensation*, arXiv:1711.07982 [quant-ph].