# Peer review of "From gauging to duality in one-dimensional quantum lattice models"

_SciPost Physics_

## Round 1 · Referee Report · Anonymous (Referee 1) · 2025-11-30

Report

The authors study the gauging of fusion-category symmetries in one-dimensional lattice systems. In this setup, gauging is specified by a choice of a haploid spherical Frobenius algebra. In the special case of an ordinary group-like symmetry, this choice determines the non-anomalous subgroup being gauged as well as the corresponding discrete torsion. The paper reviews how "gauge degrees of freedom" can be introduced in such systems, formulates an appropriate Gauss law, and constructs the corresponding gauging maps. A key contribution is the demonstration that these gauging maps are equivalent—up to constant-depth local circuits—to the duality operators developed in earlier work by some of the authors. The authors also prove a nice theorem: when the symmetry is fully gaugeable (i.e., anomaly-free), it is compatible with the existence of a short-range entangled symmetric state, matching expectations from ’t Hooft anomaly considerations.

Overall, the paper is timely and interesting within the ongoing development of categorical symmetries. The main results are reasonable and align with what one expects from other viewpoints. While Sections 1 and 2 are clearly written, I found Section 3 difficult to follow due to heavy reliance on notation and conventions from previous papers. I therefore recommend a major revision of Section 3 to make it more self-contained and accessible to a broader readership. In particular:

(1) The graphical notation introduced in Eq. (16) is dense and appears almost without any explanation.

(2) A clearer and more explicit description of the Hilbert space structure in Eq. (19) would be helpful.

(3) The relation between Section 2 and Section 3 is not transparent: Section 2 seems to assume a standard tensor-product Hilbert space, whereas Section 3 explicitly works in a constrained, non-tensor-product setting; the connection between these viewpoints should be clarified. Is section 2 a special case of section 3?

I also have several minor comments:

(a) In the fourth paragraph of the introduction, “two-dimensional” refers to two spacetime dimensions, while “one-dimensional” earlier refers to spatial dimension. It would help to maintain consistent dimensional conventions throughout.

(b) In the same paragraph, the sentence “Such an algebra object encodes both a collection of symmetry lines that are gauged as well as a way to cancel the anomaly, or discrete torsion” is misleading. Discrete torsion should not be confused with anomaly cancellation: if the anomaly vanishes, the symmetry is gaugeable, and then one may have multiple gauging choices, classified as discrete torsion. But discrete torsion (or its generalization in terms of algebra objects) doesn't cancel the anomaly.

(c) The index $i$ is used in Section 2 to label basis states of a local physical Hilbert space, but in Section 3 it labels simple objects of $\mathcal{R}$. This overloading of notation makes it harder to see how the two sections are related. I recommend distinguishing these indices and streamlining other notations.

Recommendation

Ask for major revision

---

## Editorial Decision

in_refereeing